# Identifying proteins bound to native mitotic ESC chromosomes reveals chromatin repressors are important for compaction

Dounia Djeghloul [1], Bhavik Patel[2], Holger Kramer [3], Andrew Dimond [1], Chad Whilding[4], Karen Brown[1], Anne-Céline Kohler[1], Amelie Feytout[1], Nicolas Veland [1], James Elliott[2], Tanmay A. M. Bharat [5], Abul K. Tarafder[5], Jan Löwe [6], Bee L. Ng [7], Ya Guo[1], Jacky Guy [8], Miles K. Huseyin [9], Robert J. Klose[9], Matthias Merkenschlager [1] & Amanda G. Fisher [1✉]

Epigenetic information is transmitted from mother to daughter cells through mitosis. Here, to identify factors that might play a role in conveying epigenetic memory through cell division, we report on the isolation of unfixed, native chromosomes from metaphase-arrested cells using flow cytometry and perform LC-MS/MS to identify chromosome-bound proteins. A quantitative proteomic comparison between metaphase-arrested cell lysates and chromosome-sorted samples reveals a cohort of proteins that were significantly enriched on mitotic ESC chromosomes. These include pluripotency-associated transcription factors, repressive chromatin-modifiers such as PRC2 and DNA methyl-transferases, and proteins governing chromosome architecture. Deletion of PRC2, Dnmt1/3a/3b or Mecp2 in ESCs leads to an increase in the size of individual mitotic chromosomes, consistent with de-condensation. Similar results were obtained by the experimental cleavage of cohesin. Thus, we identify chromosome-bound factors in pluripotent stem cells during mitosis and reveal that PRC2, DNA methylation and Mecp2 are required to maintain chromosome compaction.

[1] Lymphocyte Development Group, MRC London Institute of Medical Sciences, Imperial College London, Hammersmith Hospital Campus, Du Cane Road, London W12 0NN, UK. [2] Flow Cytometry Facility, MRC London Institute of Medical Sciences, Imperial College London, Hammersmith Hospital Campus, Du Cane Road, London W12 0NN, UK. [3] Biological Mass Spectrometry and Proteomics Facility, MRC London Institute of Medical Sciences, Imperial College London, Hammersmith Hospital Campus, Du Cane Road, London W12 0NN, UK. [4] Microscopy Facility, MRC London Institute of Medical Sciences, Imperial College London, Hammersmith Hospital Campus, Du Cane Road, London W12 0NN, UK. [5] Sir William Dunn School of Pathology, University of Oxford, OX1 3RE Oxford, UK. [6] MRC Laboratory of Molecular Biology, Cambridge CB2 0QH, UK. [7] Wellcome Sanger Institute, Wellcome Genome Campus, Hinxton, Cambridge CB10 1SA, UK. [8] The Wellcome Centre for Cell Biology, University of Edinburgh, Edinburgh EH9 3BH, UK. [9] Department of Biochemistry, University of Oxford, OX1 3QU Oxford, UK. ✉email: amanda.fisher@lms.mrc.ac.uk

Cell division requires genetic and epigenetic information to be accurately conveyed to daughter cells. This relies upon DNA synthesis during the S-phase of cell cycle and the subsequent segregation of this information during mitosis (M-phase). In recent years, huge progress has been made in understanding not only how chromosomal DNA is copied and segregated, but also how epigenetic information is transmitted through the cell cycle. For example, in S-phase we know that DNA methylation is normally reinstated on newly replicated DNA strands through the activity of Dnmt1, an enzyme that re-establishes methylation at hemi-methylated CpG sites[1,2]. We also know that even though histones are displaced during S-phase, histone chaperones, such as minichromosome maintenance complex (Mcm)2, ensure that parental histones are evenly allocated to the leading and lagging strands[3–6]. In addition, the epigenetic modifier polycomb repressor complex 2 (PRC2) can both methylate histone H3 at lysine 27 (through Ezh2) and recognise this mark (through Eed)[7], potentially ensuring that histone H3K27me3 is appropriately copied at newly synthesised DNA[8].

Understanding how epigenetic information is transmitted through mitosis, as newly replicated genomes condense and segregate, remains only partly understood. Progressive activation of CyclinB1-Cdk1 promotes chromosome condensation[9] so that visibly discrete individual mitotic chromosomes appear at the mitotic spindle ahead of breakdown of the nuclear envelope[10]. Although it was originally thought that most DNA sequence-specific transcription factors were actively displaced from chromosomes during mitosis[11], subsequent studies of the *Hsp70* gene promoter[12], or examining the dynamic distribution of Gata1, FoxaI and Esrrb proteins in cycling cells have revealed that many factors remain bound to mitotic chromosomes, and may occupy a subset of the genomic sites bound during interphase[13–16]. Several studies have described the dynamic changes in the repertoire of chromatin- and DNA-binding proteins, as cells transit the cell cycle[17–19]. Since the transmission of gene expression features from mother to daughter cells has been linked to DNA sequence-specific transcription factor binding through cell division[20], much attention has been focused on defining chromatin-bound mitotic factors that could activate gene expression in daughter cells following division. Some of these factors are proposed to 'bookmark' the mitotic genome, effectively marking out genes for subsequent activity[13–16,18,21–25]. In comparison, the significance of repressive chromatin modifiers that have been detected in mitotic samples[17,19,26] remains much less clear. Furthermore, although some DNA-binding factors may be retained on mitotic chromosomes through binding to their cognate motifs, other interactions may be sustained through the emergent properties of condensed mitotic chromatin[27–32].

To comprehensively evaluate the proteins that remain bound to mitotic chromosomes, we sought a high-throughput approach. As previous reports had shown that fixatives, that were intended to stabilise or cross-link mitotic chromosome preparations, can artificially displace factors from mitotic chromosomes[33–35], it was important to use unfixed chromosome samples. Prior studies indicated that native (unfixed) chromosomes could be isolated from different cell types and species directly by staining with the DNA dyes Hoechst 33258 and chromomycin A3, and sorting chromosomes by flow cytometry[36–38]. This purification step has the additional advantage over conventional approaches, as it enables a rigorous exclusion of interphase and cytoplasmic contaminants.

In this study metaphase-arrested mouse ESCs are stained with Hoechst 33258 and chromomycin A3, and flow cytometry is used to enumerate and sort specific chromosomes on the basis of AT/GC content and forward scatter (Fig. 1). Liquid chromatography-tandem mass spectrometry (LC-MS/MS) analysis of conventional metaphase-arrested ESCs, and highly enriched (flow-sorted) chromosomes, enables a catalogue of the factors present in mitotic ESCs to be compiled, where chromosome-bound factors are discriminated as being significantly enriched in chromosome-sorted fractions. Among 5888 proteins in mitotic ESC samples, ~10% (615) are significantly enriched on purified mitotic chromosomes. These include transcription factors, such as Esrrb, Sox2 and Sall4; members of the structural maintenance of chromosomes (Smc) family of proteins; heterochromatin-associated proteins and the chromatin repressors Dnmt1, Dnmt3a, Dnmt3b, Mecp2, PRC1 and PRC2. Interestingly, in ESCs that lack PRC2 activity, DNA methylation or Mecp2, mitotic chromosomes are de-condensed relative to equivalents in wild-type (WT) ESCs, consistent with these components being important for maintaining chromosome compaction. Our study describes an alternative approach for studying the properties of native mitotic chromosomes and offers a comprehensive catalogue of chromosome-bound proteins in pluripotent mouse ESCs during mitotic division.

## Results

### Isolation of native metaphase chromosomes from mouse ESCs.
We adapted a protocol used previously to isolate unfixed mitotic chromosomes[38]. Briefly, rapidly dividing cultures of mouse ESCs are arrested in metaphase using demecolcine to achieve samples where most (85–90%) cells are in M-phase as judged by propidium iodide (PI) labelling (Fig. 1a, Supplementary Fig. 1a). Condensed chromosomes are released using polyamine buffer, stained with Hoechst 33258 and chromomycin A3 as described previously[38], and examined by flow cytometry using a Becton Dickinson Influx equipped with specialised air-cooled lasers (see 'Methods' section). This approach allows individual chromosomes to be discerned and either sorted en masse (upper plot, Fig. 1a), or gated on individual chromosomes, such as chromosome X or chromosome 19 (highlighted separately in lower plot, Fig. 1a, Supplementary Fig. 1b). After sorting, the integrity of sorted chromosomes was examined and verified by optical imaging using antibody to Cenpa or using Trf1-YFP to confirm centromere, and telomere number and location (Supplementary Fig. 1c).

### Analysis of proteins bound to metaphase ESC chromosomes.
To determine the proteins bound to native (unfixed) metaphase ESC chromosomes, we performed a proteomic analysis using LC-MS/MS and analysed the data using the label-free quantification (LFQ) algorithm within the MaxQuant software platform (Fig. 1a–c). In these experiments, we compare samples containing equivalent numbers ($10^7$) of ESC metaphase chromosomes before (mitotic lysate pellet) and after chromosome sorting, in three biological replicates (Supplementary Fig. 1d shows pairwise comparisons between replicates). We identify 5888 proteins in mitotic lysates, of which 5436 are identified with two or more razor or unique peptides per protein at a 1% false discovery rate (FDR). Our rationale is that chromosome-bound factors should be enriched in sorted samples, while factors that are not chromosome-associated would be depleted. In chromosome-sorted fractions, we detect 3749 proteins that are either significantly enriched as compared to pelleted mitotic lysates (615, red), are depleted (1548, blue), or showed no statistical difference between the two (1354, black; Fig. 1b, c). Hierarchical clustering of these proteomics datasets is shown in Supplementary Fig. 1e, and GO term annotation of enriched and depleted candidates is shown in Supplementary Fig. 1f. We note that the number and identity of proteins detected is broadly similar to those identified in previous studies of mitotic human and chicken

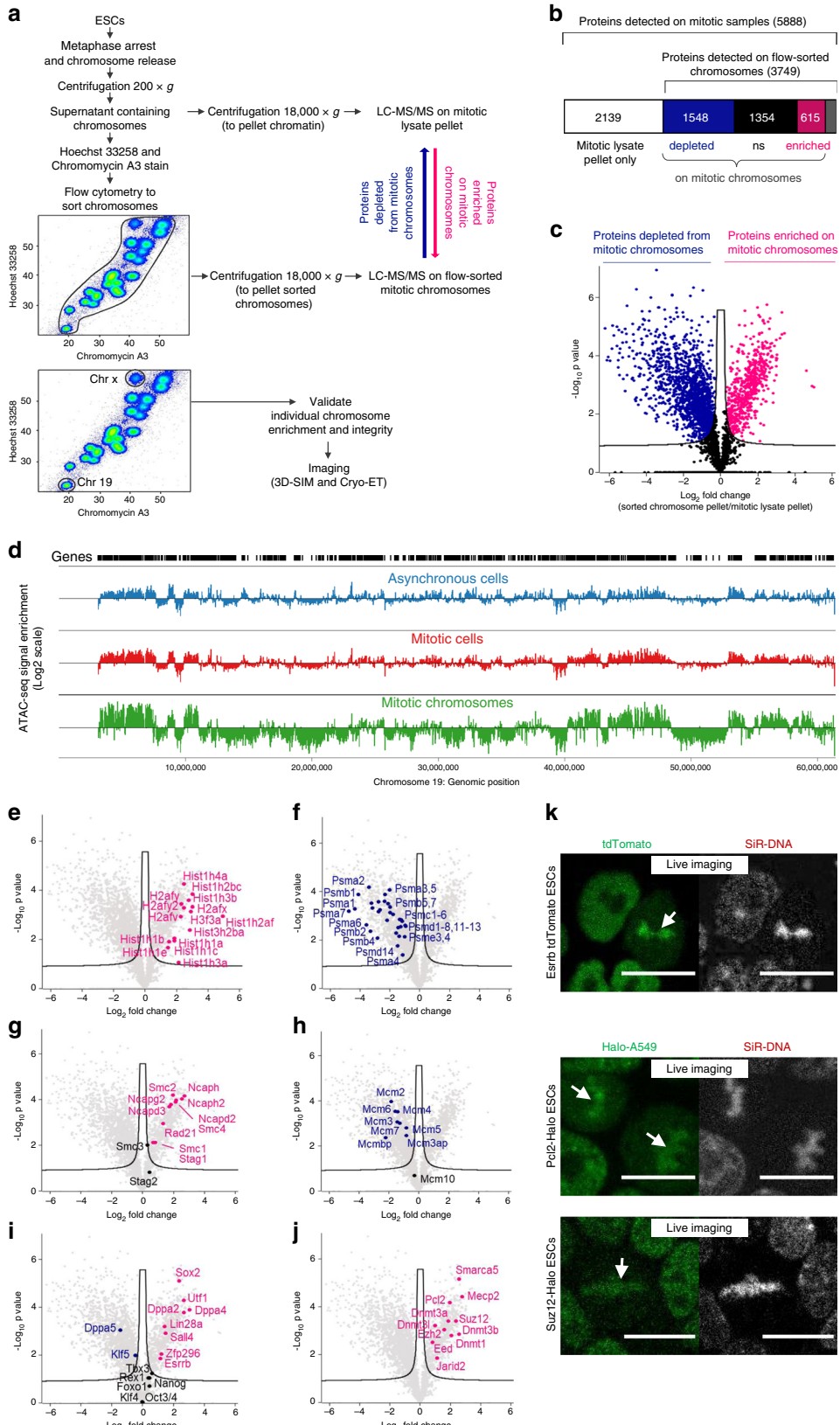

cells[17,19] (Supplementary Fig. 1g), suggesting that our method of isolating native mitotic chromosomes does not incur any large-scale loss of chromatin-bound proteins. Furthermore, chromatin accessibility is known to be largely maintained in mitosis[16,33,39], and ATAC-seq analysis demonstrates that global chromatin accessibility

patterns are broadly preserved in flow-sorted mitotic chromosomes (Fig. 1d).

Proteomic comparisons between sorted- and conventional metaphase-arrested samples (Fig. 1e–j) reveal a statistical enrichment in histones, Smc-associated proteins and many sequence-

**Fig. 1 Proteins bound to ESC-derived metaphase chromosomes. a** Scheme of experimental strategy used to isolate native metaphase chromosomes from ESCs and identify proteins bound to mitotic chromatin. Hoechst 33258 and chromomycin A3 bivariate karyotype was assessed by flow cytometry, and the gates used to sort all chromosomes, chromosome 19 or the X chromosome are indicated. Proteomic analysis was performed using LC-MS/MS on total mitotic cell lysate pellet, or on flow-purified chromosomes, to identify proteins bound to native metaphase chromosomes. **b** Diagram showing the number of proteins identified by proteomic analysis in mitotic lysate pellet and chromosome-sorted samples. **c** Volcano plot of proteins detected as being significantly enriched or depleted on sorted chromosomes relative to mitotic lysate pellet (unpaired two tailed Student's *t*-test, permutation-based FDR < 0.01, *n* = 3 independent experiments each measured in duplicate, see 'Methods' section for details). Proteins were plotted as Log2 fold change (LFQ intensity of sorted chromosome pellet /LFQ intensity of mitotic lysate pellet) and significance (−Log10 *p*) using Perseus software. **d** Chromatin accessibility profile across chromosome 19 for asynchronous and mitotic cells, and flow-sorted chromosomes, shown as Log2 enrichment of ATAC-seq signal. **e–j** Volcano plots as in **c**, highlighting **e** histones, **f** components of the proteasome, **g** Smc-associated proteins, **h** DNA replication machinery, **i** pluripotency-associated transcription factors or **j** chromatin repressors that are enriched (red), depleted (blue) or not significantly enriched (ns, black) on ESC mitotic chromosomes versus mitotic lysate pellet. **k** Localisation of Esrrb, Pcl2 and Suz12 fusion proteins (green, left panels) to mitotic chromatin in live ESCs cultured with SiR-DNA (grey, right panels). Arrows show Esrrb, Pcl2 and Suz12 localisation to mitotic chromatin. Scale bars = 14 µm. Images are representative of three independent experiments.

specific DNA-binding factors in chromosome-sorted samples (Fig. 1e, g, i, respectively), consistent with these factors remaining intimately associated with chromosomes in mitosis. In contrast, as expected, components of the proteasome (Fig. 1f) and members of the Mcm (known to dissociate after S-phase, Fig. 1h), are depleted from sorted chromosome fractions, as are cytoplasmic, organelle- and membrane-associated proteins (Supplementary Fig. 1f). Several transcription factors known to regulate ESC pluripotency and differentiation show differential associations with mitotic chromosomes (Fig. 1i). For example, Utf1, a transcription factor required for the proper differentiation of embryonic carcinoma cells and ESCs[40], is enriched after chromosome sorting, while Nanog, Oct4 (Pou5f1) and Klf4, although detected, show no significant enrichment in sorted chromosome samples. Esrrb and Sox2, two previously characterised bookmarking factors[15,16,22,33], are both significantly enriched in chromosome-sorted samples, while Dppa5 and Klf5 are depleted. Consistent with Esrrb genomic bookmarking being preserved on native flow-sorted mitotic chromosomes, ATAC-seq analysis of binding sites reported to be preserved or lost in mitosis[16] shows a characteristic retention or loss of accessibility, consistent with prior studies (Supplementary Fig. 1h). Interestingly, proteins associated with gene repression and heterochromatin formation, including the histone methyl-transferases Suv39h1 and Suv39h2, PRC1 and PRC2 components Suz12, Eed, Ezh2, Jarid2 and Pcl2, are significantly enriched in sorted chromosome samples (Fig. 1j, Supplementary Fig. 1f, Supplementary Data File 1). The DNA methyl-transferases Dnmt1, Dnmt3a and Dnmt3b are also co-enriched after chromosome sorting, suggesting that most of the chromatin machinery required to sustain H3K9me3, H3K27me3 and 5mC marking of the genome remain bound to ESC chromosomes during mitosis. Likewise, the methyl-CpG-binding protein Mecp2 and the SWI/SNF-related protein Smarca5 also show significant enrichment in sorted chromosome samples (Fig. 1j). Smarca5 is an important chromatin remodelling factor that is involved in establishing regularly spaced nucleosomes, critical for DNA replication and repair, and associated with both positive and negative transcriptional outcomes. Smarca5 has also been shown to interact with Rad21 (ref. [41]), and co-enrichment of Rad21, Smc1, Smc2 and other Smc-associated proteins is also evident in chromosome-sorted samples (Fig. 1g). Antibody labelling was performed for selected factors, confirming their relative enrichment (Sox2 and Rad21) or lack of enrichment (Nanog and Oct4) on flow-sorted chromosomes (Supplementary Fig. 1i). To validate the mitotic binding of candidates identified by proteomic analysis of native isolated chromosomes, we performed live cell imaging in ESCs engineered to express tdTomato or Halo-tag fusion proteins. As shown in Fig. 1k, and the accompanying video (Supplementary Movie 1), PRC2 components Suz12 and Pcl2 (green) show a dispersed nuclear distribution in interphase cells, but

colocalise with chromosomal DNA (labelled with SiR, greyscale) during mitosis (arrows). This dynamic labelling profile is similar to that observed for Esrrb, a previously reported bookmarking factor in ESCs[16].

## DNMTs and PRC2 activity keep mitotic chromosomes compact.
To evaluate the functional relevance of PRC2 and DNMTs on mitotic chromosomes, we examined native chromosomes isolated form ESCs that lacked either DNA methylation[42] or PRC2 activity[43]. While the overall distribution of mitotic chromosomes isolated from WT and mutant (*Dnmt1,3a,3b*[−/−] or *Eed*[−/−]) ESCs appears similar (Fig. 2a), closer inspection reveals differences in their size and shape. To accurately measure this, we separately purified two representative mitotic chromosomes (19 and X) from WT and mutant ESCs, using Hoechst 33258 and chromomycin A3 staining and flow sorting, as described previously (Fig. 1a). DNA FISH with mouse chromosome 19- or X-specific paints (Supplementary Fig. 2a) confirms 99–100% sample purity. Individual chromosomes were measured by microscopy using standard imaging software to determine the total chromosome area and estimate the size of DAPI-bright pericentric domains (Supplementary Fig. 2b). These analyses indicate that mitotic 19 and X chromosomes isolated from *Dnmt1,3a,3b*[−/−] (23.4 ± 4.1, 43.6 ± 6.9 µm² respectively) or *Eed*[−/−] (24.3 ± 4.5, 42.2 ± 4.7 µm², respectively) ESCs are significantly larger than equivalent chromosomes isolated from WT ESCs (18.4 ± 3, 35.8 ± 5.3 µm², respectively), while those from a Sox2-deficent ESC line[44] are of comparable size and shape (Fig. 2b, c). Centromere size is also increased in mitotic samples that lack DNA methylation or PRC2 activity (Fig. 2b, c). Although *Dnmt1,3a,3b*[−/−] ESCs lack 5mC and have elevated H3K27me3 at pericentric domains in interphase as compared to WT[45] (Supplementary Fig. 2c, d), we do not observe similar increases in H3K27me3 on these mitotic chromosomes (Supplementary Fig. 2e, red). Instead these chromosomes have reduced H3K9me3 levels, particularly at the centromeres (Supplementary Fig. 2e, middle panel, green). These data suggest that a lack of either H3K27me3 or H3K9me3 can impair chromosome compaction.

To confirm that DNA methylation and PRC2 are important for maintaining chromosome compaction, we examined conventional metaphase spreads from WT and mutant ESCs, using DNA FISH to separately identify and measure mouse chromosomes 19 and pericentric γ-satellite repeats (illustrated in Fig. 2d, left). We observe an increase in the overall size of chromosome 19 in metaphase spreads from *Dnmt1,3a,3b*[−/−] and *Eed*[−/−] ESCs as compared to WT samples and, consistent with previous results, see no appreciable differences in *Sox2*[−/−] mutants. In *Dnmt1,3a,3b*[−/−] samples we also note increased chromosome 19 centromere size, although this is not significant for *Eed*[−/−]

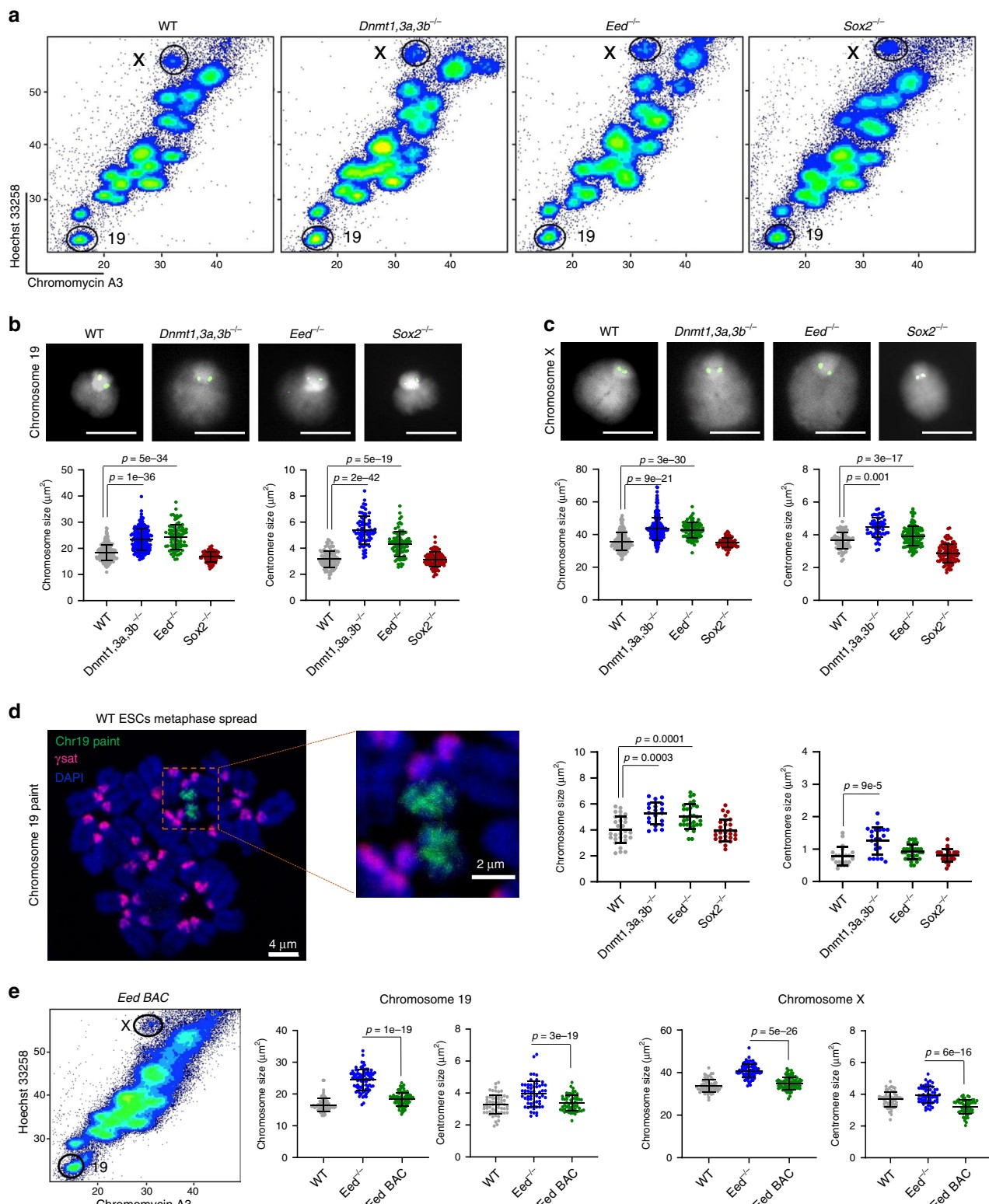

samples. Together these experiments suggest that DNA methylation and PRC2 activity have roles in ensuring efficient chromosome compaction in mitosis. To exclude that the chromosome decompaction seen in PRC2-deficient ESCs is due to any inadvertent secondary effects in mutant $Eed^{-/-}$ cells, we examined mitotic chromosomes from $Eed^{-/-}$ ESCs, in which PRC2 activity and H3K27me3 had been restored by transfection with a BAC clone that contained $Eed$ (clone B1.3 BAC)[43]. As shown in Fig. 2e, metaphase chromosomes 19 and X isolated from these Eed-rescued ESCs ($Eed\ BAC$) are similar in size and shape to equivalent chromosomes from WT ESCs. These results confirm that the decompaction of mitotic ESC chromosomes seen in the absence of Eed is fully reversed by restoring PRC2 activity.

**Fig. 2 Increased size of ESC metaphase chromosomes that lack DNA methylation or PRC2 activity. a** Flow karyotype of mitotic chromosomes isolated from WT ESCs or mutant ESCs that lack Dnmt1/3a/3b, Eed or Sox2. Gates used to isolate chromosomes 19 or X are indicated. Images are representative of three independent experiments. **b, c** Representative images of mitotic chromosomes 19 (**b**) and X (**c**) from different ESCs are shown, where DAPI stain (grey) and Cenpa label (green) indicate the chromosome body and centromere, respectively. Scale bars = 5 µm. Chromosome and centromere sizes were calculated for each ESC line by measuring individual chromosomes (chromosome 19: $n = 217, 182, 100$ and $101$; chromosome X: $n = 189, 201, 99$ and $98$) and centromeres (chromosome 19: $n = 100, 82, 90$ and $100$; chromosome X: $n = 76, 71, 114$ and $98$) over three independent experiments, mean ± SD are shown. $P$-values of statistically significant increases, measured by unpaired two tailed Student's $t$-tests, are indicated. **d** Representative image of ESC metaphase spread stained with chromosome 19 painting probe (green), gamma satellite probe (red) and DAPI (blue). Scale bars = 4 µm and 2 µm for the metaphase spread and zoom-in images, respectively. Chromosome and centromere sizes of chromosome 19 were calculated by measuring metaphase spreads of WT ($n = 29$), Dnmt1,3a,3b$^{-/-}$ ($n = 20$), Eed$^{-/-}$ ($n = 31$) and Sox2$^{-/-}$ ($n = 28$) ESCs over three independent experiments, mean ± SD are shown. $P$-values of statistically significant increases, measured by unpaired two tailed Student's $t$-tests, are indicated. **e** Flow karyotype and mitotic chromosome sizes of Eed$^{-/-}$ ESCs before and after restoring Eed expression (Eed BAC). Chromosome and centromere sizes were measured for each ESC line by measuring individual chromosomes (chromosome 19: $n = 137, 86$ and $80$; chromosome X: $n = 130, 91$ and $81$) and centromeres (chromosome 19: $n = 100, 90$ and $102$; chromosome X: $n = 84, 113$ and $100$) over three independent experiments, mean ± SD values are shown. $P$-values of statistically significant decreases, measured by unpaired two tailed Student's $t$-tests, are indicated. **b–e** Source data are provided as a Source data file.

**Mecp2 contributes to mitotic compaction of autosomes in ESCs.** The methyl-binding protein Mecp2 has been implicated in regulating chromatin architecture at a range of different levels, from the juxtaposition of nucleosome arrays, to condensing pericentric heterochromatin[46]. Mecp2 binds methylated CpG di-nucleotides, but has also been reported to interact with other partners independent of its methy-CpG-binding activity[47]. Proteomic data shown in Fig. 1j indicate that Mecp2 is bound to mitotic chromosomes. To verify this, we examined Mecp2 distribution in live cells, using a previously generated Mecp2-eGFP ESC line[48]. As illustrated in Fig. 3a and the associated video (Supplementary Movie 2), Mecp2 (green) co-localises with condensed chromosomes (labelled with SiR-DNA, greyscale) throughout mitosis. To test whether the mitotic chromosome decompaction observed in Dnmt1,3a,3b$^{-/-}$ ESCs could be at least partially due to abrogated interactions with methyl-binding proteins, such as Mecp2 (rather than reduced methylation per se), we examined mitotic chromosomes from ESCs, in which Mecp2 had been deleted[49]. Comparing flow-sorted metaphase chromosomes 19, 3 and X isolated from parental male Mecp2-expressing (Mecp2$^{lox/y}$) and Mecp2-deficient (Mecp2$^{-/y}$) ESCs (Fig. 3b), we observe significant increases in the sizes of both autosomes in the absence of Mecp2, but no change in the size of the active X chromosome (Fig. 3c). Although CpG methylation is found on active and inactive X chromosomes, analysis of published ChIP-seq data[50,51] indicates that Mecp2 is less abundant on the X chromosome than on autosomes (Supplementary Fig. 3a, b). Mecp2 depletion also results in chromosome 19 and chromosome 3 centromere decompaction (Fig. 3c). This is associated with less pericentric H3K9me3 and a reduction in H3K27me3 labelling on mitotic chromosomes (Supplementary Fig. 3c). To confirm the importance of Mecp2 for autosome compaction during mitosis, we analysed conventional metaphase spreads prepared from WT and Mecp2-deficient ESCs, using DNA FISH to identify chromosome 19 and pericentric γ-satellite repeats. These analyses, shown in Fig. 3d, confirm that Mecp2 depletion results in an increase in chromosome 19 size.

**Mitotic chromosome size depends upon differentiation state.** PRC2 activity and DNA methylation are required for the successful differentiation of ESCs, but not for ESC self-renewal or pluripotency (reviewed in refs. [52,53]). We therefore asked whether mitotic chromosomes isolated from differentiated cells and ESCs are similar. To examine this, we isolated individual chromosomes from metaphase-arrested mouse ESCs (Fig. 4a), pre-B cells (Fig. 4b), mouse cardiomyocyte HL-1 cells (Fig. 4c) and primary embryonic fibroblasts (Fig. 4d). Although the success of metaphase arrest varies between the different cell types (45–90%), by applying our flow

cytometry-based approach we are able to isolate and purify mitotic chromosomes, irrespective of differences arising from cell cycle synchronisation or karyotype complexity. Using chromosomes 19 and X as representatives, we find that native mitotic chromosomes isolated from differentiated pre-B cells, cardiomyocytes and fibroblasts are smaller than equivalents isolated from ESCs (Fig. 4e, f). This is confirmed by FISH analysis of conventionally prepared metaphase spreads (Fig. 4g). Interestingly, although the sizes of mitotic chromosomes 19 and X are reproducibly larger in ESCs than in each of the differentiated cell types examined, no differences in the sizes of centromeres between each of the cell types is observed (right hand panels of Fig. 4e–g). This is consistent with constitutive heterochromatin domains being relatively well conserved across different cell types and differentiation stages[52]. As differences in mitotic chromosome size could reflect constraints imposed earlier in the cell cycle, for example, by the size of nuclei, we measured the diameter of G1- and G2-phase nuclei in each of the different cell types (Supplementary Fig. 4a). Although nuclear size varies between different cell types and mutant ESC lines (Supplementary Fig. 4b, c, respectively), no correspondence between chromosome size and nuclear dimensions is evident. This observation effectively rules out a simple passive correlation between mitotic chromosome size and nuclear size, at least for the examples studied here. As a caveat, it is conceivable that differences in the arrest times required to obtain metaphase samples of different cell types could confound estimates of chromosome size in these samples.

**Isolated chromosomes are sensitive to in situ cohesin cleavage.** A major potential advantage of purifying native chromosomes from cells in mitosis is that these unfixed chromosomes could be used as a substrate to examine the impacts of experimental perturbations applied in situ. As an example to test this, we asked whether experimental cleavage of cohesin complexes significantly alters the structure of isolated metaphase chromosomes. Cohesin complexes are composed of Smc1, Smc3, the kleisin Scc1, and one of three auxiliary subunits[54] and are required to keep sister chromatids together from DNA replication until mitosis, as well having roles in interphase genome organisation, gene transcription and DNA repair[55–59]. Cohesin complexes are dynamically regulated during the cell cycle. While the majority of cohesin is thought to dissociate from chromosome arms during prophase, recent studies have suggested that residual cohesin is critical for retaining elongating RNA polymerase II at centromere domains[60]. Some cohesin remains bound to centromeres until anaphase when separase cleaves the kleisin subunit[54,61,62]. Consistent with this, cohesin components are detected within the metaphase proteome, both by ourselves (Fig. 1g), and others[17,19]. To investigate the impact of cohesin cleavage on sorted native

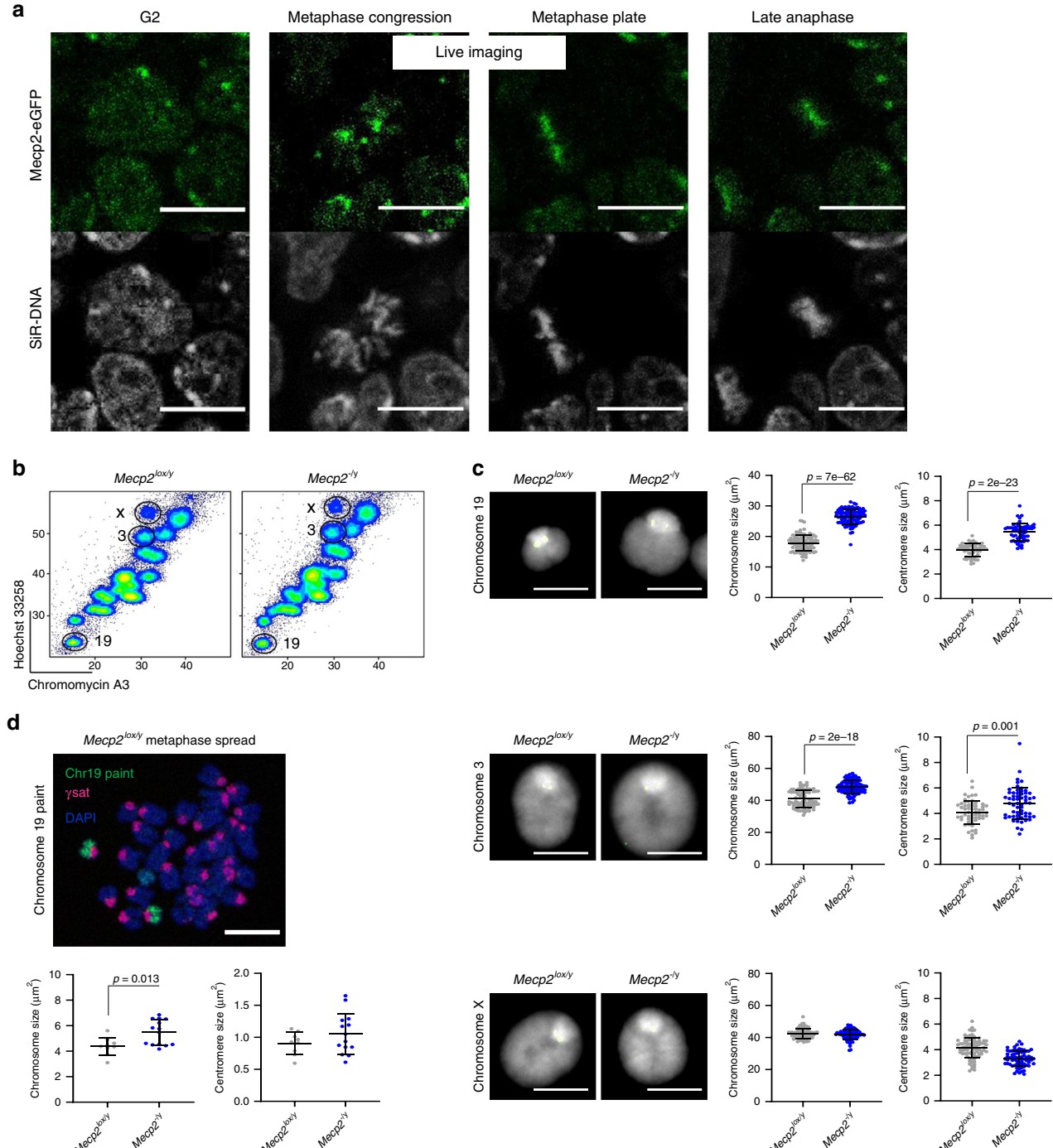

**Fig. 3 Increased size of mitotic chromosomes in ESCs lacking Mecp2. a** Mecp2 association with chromatin throughout mitosis using live cell imaging of Mecp2-eGFP fusion protein in ESCs. Selected time frames from the same dividing cell are shown. Scale bar = 14 μm. Images are representative of three independent experiments. **b** Flow karyotype of mitotic chromosomes isolated from *Mecp2*^lox/y or *Mecp*^−/y ESCs. Gates used to isolate chromosomes 19, 3 or X are indicated. Images are representative of three independent experiments. **c** Representative images of mitotic chromosomes 19, 3 and X from Mecp2^lox/y and *Mecp2*^−/y ESCs are shown, where DAPI stain (grey) and Cenpa label (green) indicate the chromosome body and centromere, respectively, scale bars = 5 μm. Chromosome and centromere sizes were calculated for each ESC line by measuring individual chromosomes (chromosome 19: $n = 100$ and 100; chromosome 3: $n = 80$ and 100; chromosome X: $n = 100$ and 100) and centromeres (chromosome 19: $n = 55$ and 60; chromosome 3: $n = 60$ and 60; chromosome X: $n = 80$ and 70) over three independent experiments, mean ± SD are shown. **d** Representative image of *Mecp2*^lox/y ESC metaphase spread stained with chromosome 19 painting probe (green), gamma satellite probe (γsat, pink) and DAPI (blue), scale bar = 4 μm. Chromosome and centromere sizes of chromosome 19 were calculated by measuring metaphase spreads of *Mecp2*^lox/y ($n = 8$) or *Mecp2*^−/y ($n = 14$) ESCs, mean ± SD are shown. **c**, **d** P-values of statistically significant increases, measured by unpaired two tailed Student's t-tests, are indicated. Source data are provided as a Source data file.

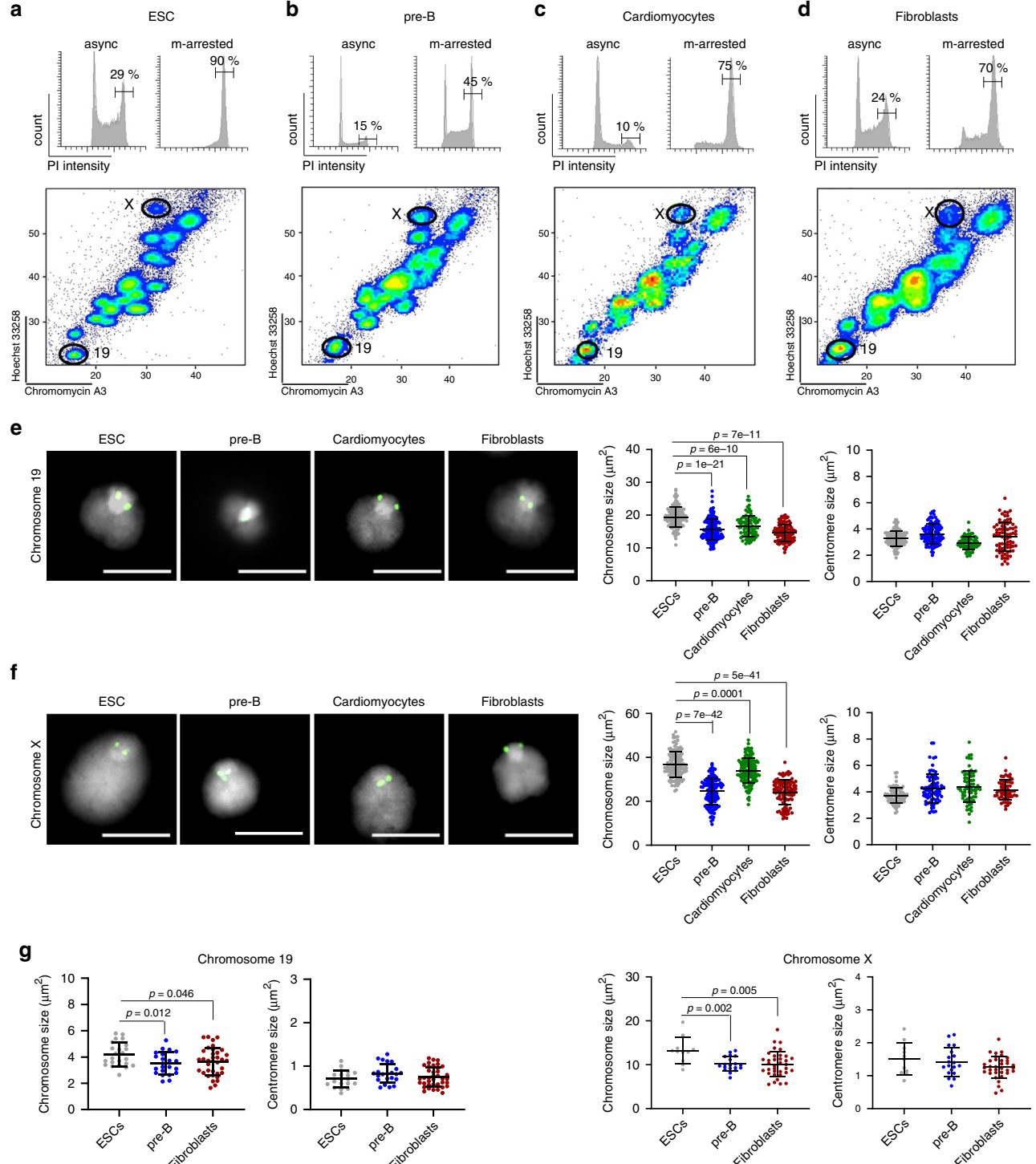

**Fig. 4 Native mitotic ESC chromosomes are larger than equivalents isolated from differentiated cells. a–d** Cell cycle profiles of mouse ESCs (**a**), pre-B cells (**b**), cardiomyocytes (**c**) and embryonic fibroblasts (**d**) were determined by staining with propidium iodide. Left panel shows asynchronized (async) cells, right panel shows samples 6–12 h after treatment with demecolcine (m-arrested), where values indicate the percentage cells in G2/M stage. Lower panel shows flow karyotype of demecolcine-treated cells and the gates used to isolate chromosomes 19 and X. Images are representative of three independent experiments. **e, f** Representative images of native mitotic chromosomes 19 (**e**) and X (**f**) isolated from mouse ESCs, pre-B cells, cardiomyocytes (HL-1) and embryonic fibroblasts. DAPI stain (light grey) and Cenpa (green) labelling are shown, scale bars = 5 μm. Chromosome and centromere sizes were determined for each cell type by measuring individual chromosomes (chromosome 19: $n$ = 137, 149, 88 and 106; chromosome X: $n$ = 129, 132, 134 and 103) and centromeres (chromosome 19: $n$ = 119, 100, 72 and 83; chromosome X: $n$ = 81, 81, 70 and 60) over three independent experiments, mean ± SD are indicated. **g** Chromosome and centromere sizes of chromosome 19 (left panel) and chromosome X (right panel) were calculated by measuring metaphase spreads of ESCs ($n$ = 23), pre-B cells ($n$ = 22) and embryonic fibroblasts ($n$ = 36), mean ± SD are indicated. **e–g** P-values of statistically significant decreases, measured by unpaired two tailed Student's $t$-tests, are indicated. Source data are provided as a Source data file.

metaphase chromosomes, we used a model pre-B cell line that expresses a cleavable form of Rad21 (Rad21-TEV-myc)[63,64], as illustrated in Fig. 5a. In these pre-B cells, Rad21 binding is detected on metaphase samples (Supplementary Fig. 5a, Rad21 and Myc labelling, green), as well as around the centromeres of sorted native mitotic chromosomes (Supplementary Fig. 5b). We performed Hi-C analysis of flow-sorted mitotic chromosomes from pre-B cells to confirm that the 3D chromosome contacts present during interphase (Supplementary Fig. 5c, upper panels) are lost from mitotic chromosome samples (Supplementary Fig. 5c, lower panels), consistent with previous reports[65,66]. We then isolated native mitotic chromosomes from WT and $Rad21^{Tev/Tev}$ pre-B cell lines (Supplementary Fig. 5d), and examined the impact of cohesin cleavage induced by TEV protease (Fig. 5a), using advanced optical microscopy or cryo-electron tomography (cryo-ET; Fig. 5b shows the experimental design). TEV treatment results in efficient cleavage of Rad21-Tev, as verified by Myc immunofluorescence labelling (Fig. 5c) and western blotting (Supplementary Fig. 5e). TEV-induced cohesin cleavage results in a significant increase in the size of mitotic chromosome 19, compared with either untreated (−TEV) or TEV-treated chromosomes derived from WT pre-B cells (Fig. 5d). Increased mitotic chromosome 19 size is accompanied by a de-condensation at DAPI-bright pericentric domains (arrowed). Further cryo-ET analysis of two independent experiments (Fig. 5e) confirms that TEV-induced cohesin cleavage results in an increase in the size of mitotic ($Rad21^{Tev/Tev}$) chromosome 19, and a widespread de-condensation is evident in representative 3D images (Supplementary Movies 3 and 4).

## Discussion

The isolation and purification of metaphase chromosomes by flow cytometry offers a different approach for studying chromosome structure and function in mitosis. By combining this approach with quantitative LC-MS/MS analysis, we have been able to characterise a large repertoire of proteins that remain bound to unfixed mitotic chromosomes in pluripotent ESCs. These included proteins with established roles in chromatin organisation, DNA and nucleosome packaging, cell cycle and chromosome architecture and function. Three subsets of transcription factors relevant for pluripotency were discerned that showed either a significant enrichment on sorted metaphase chromosomes versus lysates (such as Esrrb, Sox2, Utrf1, Dppa4, Dppa2 and Sall4), were depleted (Dppa5 and Klf5) or were similarly represented in both (Oct4, Nanog and Klf4). The first group includes Esrrb and Sox2, transcription factors that have previously been shown to bind to chromosomes throughout mitosis and implicated in mitotic bookmarking in ESCs[15,16,22], while the latter two groups comprise candidates that may either be evicted from condensing chromosomes, or be in dynamic flux, appearing to be similarly distributed in mitotic lysates and chromosome samples. Importantly, we have shown that metaphase chromosomes isolated by flow cytometry retain the genome-wide chromatin accessibility features that characterise mitosis in ESCs[16], but lack interphase and cytoplasmic contaminants, that may have confounded similar past studies.

Proteomic analyses revealed that many of the core component of PRC1 (Rnf2, Pcgf6, Cbx2 and Phc1) and PRC2 (Eed, Ezh2, Suz12, Pcl2 and Jarid2), that are responsible for catalysing histone H2AK119 mono-ubiquitination and histone H3K27 tri-methylation, respectively[67], were enriched on metaphase chromosomes in ESCs. Our results were obtained using unfixed but highly purified metaphase ESCs and our data resemble those published in a recent study of chromatin-bound changes through the cell cycle of human glioblastoma T98G cells[19], which showed chromatin repressor

complexes remaining bound throughout mitosis. Prior studies in *Drosophila* embryos, as well as in human primary cells, suggested that polycomb group proteins, such as PC, PH, PSC and BMI1 dissociate from condensing chromosomes in prophase to metaphase[68,69]. However, studies of living rather than fixed samples have suggested that although GFP-tagged PC fusion proteins were depleted in mitosis relative to interphase, some PC complexes remain bound through cell division[70]. Here, using similar live cell imaging approaches, we have confirmed that PRC2 complexes remain bound to chromosomes throughout mitosis in ESCs. Our proteomic data also showed that Dnmt1, Dnmt3a, Dnmt3b and methyl-CpG-binding protein Mecp2, were enriched on metaphase ESC chromosomes, a result that was confirmed in live cell imaging experiments using eGFP-tagged Mecp2 fusion proteins. The observation that PRCs and DNA methylation machinery remain chromosome-associated throughout mitosis is intriguing, and is consistent with the possibility that mitotic memory is conveyed both by repressive chromatin states, as well as factors that bookmark the genome to activate and enhance gene expression[71].

To assess the functional impacts of chromatin repressors on chromosome structure in mitosis, we examined chromosomes from ESCs that lacked PRC2 activity, DNA methylation or Mecp2, compared with WT ESCs or cells that lacked the transcription factor Sox2. We showed that genome-wide loss of DNA methylation resulted in chromosomes and centromeres that were larger and less compact than equivalent mitotic chromosomes in WT or $Sox2^{-/-}$ ESCs. At first sight, this is surprising since in interphase reduced DNA methylation is reported to affect nuclear organisation, histone modifications and linker histone binding, but does not directly alter chromatin compaction[72]. Furthermore, at the nucleosome level, in vitro studies have indicated that DNA methylation alone does not induce chromatin compaction[73,74]. This suggests that the decompaction of metaphase chromosomes seen in $Dnmt1,3a,3b^{-/-}$ ESCs most likely stems from secondary changes that serve to relax the higher order structure of chromatin[75], perhaps through altered histone H1 binding or an impaired recruitment of Mecp2. Mecp2 is a multifunctional protein that can bind both methylated and un-methylated DNA, can compete with histone H1, and has been shown to directly mediate nucleosome oligomerisation and compaction in vitro[46]. Consistent with the reduced chromosome compaction seen in $Dnmt1,3a,3b^{-/-}$ mutants being attributable to a failure to recruit DNA-binding proteins, rather than a lack of 5mC per se, we saw impaired mitotic chromosome compaction in $Mecp2$ mutants, where DNA methylation is preserved. Mitotic chromosomes from ESCs that lacked PRC2 activity and H3K27me3 were also much less compact than equivalents from WT or Sox2 mutants. Importantly, mitotic chromosome compaction could be rescued in mutants by restoring PRC2 activity. These results, taken together, highlight a previously unrecognised role for chromatin repressors in maintaining mitotic chromosome structure.

Our results also indicate that the size of individual metaphase chromosomes differs between different cell types. Native mitotic chromosomes purified from ESCs were much less condensed than equivalents isolated from either lymphocytes, cardiomyocytes or fibroblasts. The idea that chromosomes of pluripotent ESCs might be more 'loosely packed' than somatic equivalents is consistent with previous studies in interphase, in which electron spectroscopic imaging (ESI) revealed that ESCs and cells of the mouse early epiblast (E3.5) lack the compact chromatin domains that characterise differentiated lymphocytes, liver and kidney cells[76]. ESI studies have also shown that at later stages of development (E5.5) epiblast cells lose the dispersed 10 nm chromatin fibres that are so-called architectural hallmarks of pluripotency[77]. Previous studies in embryos from *Caenorhabditis elegans* and in different *Xenopus* species have suggested that mitotic chromosome size scales with

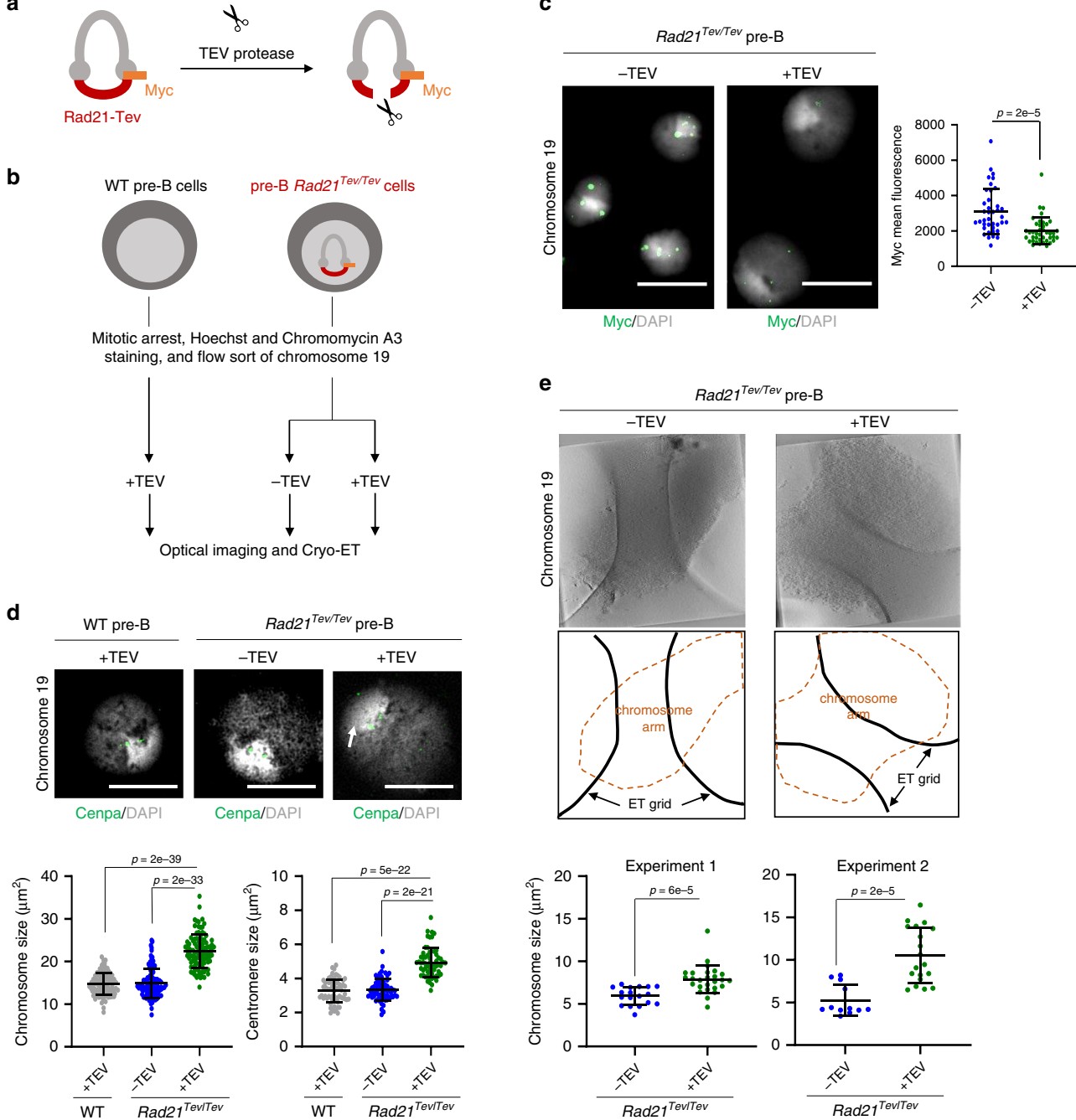

**Fig. 5 Experimentally induced cleavage of cohesin alters flow-sorted mitotic chromosome size. a** Experimental strategy used to cleave cohesin using TEV protease; illustrated is a cohesin ring containing Smc1, 3 and Rad21-Tev-Myc. **b** Scheme used to isolate and image mitotic chromosomes from WT and Rad21-Tev-myc ($Rad21^{Tev/Tev}$) pre-B cells. Mitotic chromosomes from WT pre-B cells or from $Rad21^{Tev/Tev}$ pre-B cells were purified by flow cytometry, and incubated with (+) or without (−) TEV protease. **c** Myc labelling (green) of $Rad21^{Tev/Tev}$ purified chromosome 19 shows reduced Myc levels after treatment with TEV protease (images left, and quantified by intensity, right). Scale bar = 5 μm, n = 38 chromosomes for −TEV and n = 40 chromosomes for +TEV, mean intensity values ± SD are shown. **d** Representative super-resolution SIM images of purified mitotic chromosome 19 isolated from WT or $Rad21^{Tev/Tev}$ pre-B cells, treated with TEV in situ (+TEV) or with buffer alone (−TEV). Scale bars = 2.86 μm. Chromosome and centromere sizes were determined for each condition by measuring individual chromosomes (n = 101, 101 and 101) and centromeres (n = 65, 64 and 61), values indicate mean ± SD. **e** Representative slices through cryo-electron tomograms (Cryo-ET) of chromosome 19 isolated from $Rad21^{Tev/Tev}$ pre-B cells and treated with TEV in situ (+TEV) or with buffer alone (−TEV) (top panel) and Cryo-ET image explanation (middle panel). Graphs show chromosome size, calculated as area measurements from 2D electron microscopy images, mean ± SD are indicated. Values from two independent experiments are shown, n = 20 and 28 chromosomes for experiment 1 and n = 14 and 21 chromosomes for experiment 2, for −TEV and +TEV respectively. **c–e** P-values were calculated using an unpaired two tailed Student's t-test. Source data are provided as a Source data file.

nuclear size, or RanGTP levels regulated by the chromatin-associated factor RCC1 (guanine nucleotide exchange factor)[78–80]. In the studies shown here, a correlation between nuclear size and mitotic chromosome size across different cell types was not evident. While future studies will be required to determine why mitotic chromosomes in ESCs are less condensed than those of their differentiated counterparts, we note that several candidates implicated in regulating mitotic chromosome compaction, including Cenpa, the DNA-decatenating enzyme topoisomerase II, cohesin and condensin, core and linker histones[81], were detected within the proteome of sorted mitotic ESC chromosomes.

Our ability to purify and compare individual native mitotic chromosomes isolated from different cell types, combined with appropriate genetic and biochemical tools that enable inducible protein cleavage or degradation, means that it is possible to examine the impact of specific proteins on mitotic chromosome structure. In this regard, we have demonstrated that dynamic cleavage of cohesin complexes on metaphase chromosomes in situ prompts a widespread chromosomal de-condensation that includes, but extends beyond, centromeric domains. This result indicates that mitotic chromosomes isolated in their native state remain sensitive to regulators of chromosome architecture, a finding that should enable the mechanisms of chromosome condensation and de-condensation to be more closely observed and interrogated at the molecular and structural level.

## Methods

**Cells.** Mouse ESCs used in this study were WT E14Tg2a[44], Sox2−/− (clone 2O5)[44], Dnmt1,3a,3b −/−[42], Eed−/− (clone B1.3), rescued Eed null (Eed BAC)[43], floxed Mecp2 ESC clones (Mecp2lox/y and Mecp2−/y) and Mecp2-eGFP (gift from Jacky Guy), Esrrb-tdTomato (gift from Nicola Festuccia), and Pcl2-halo and Suz12-halo-tagged ESC lines (gift from Robert J. Klose). ESCs were cultured on 0.1% gelatin-coated plates. Cells were grown in KO-DMEM medium supplemented with 15% FCS, non-essential amino acids, L-glutamine, 2-mercaptoethanol, antibiotics and 1000 U ml−1 of leukaemia inhibitory factor. Abelson-transformed pre-B cell lines (WT and Rad21Tev/Tev pre-B cells) were cultured in IMDM medium supplemented with 10% FCS, 2 mM L-glutamine, antibiotics, and 50 μM 2-mercaptoethanol and non-essential amino acids. These cells were previously derived in our lab from transgenic Rad21-Tev-Myc mice[63]. The Rad21Tev/Tev pre-B cell line expresses a modified and functional Rad21 protein containing three Tev cleavage sites within the flexible polypeptide connecting the N-terminal and C-terminal domains. The presence of a Myc tag allows us to follow the cleavage and the chromosomal localisation of Rad21 with an anti-Myc antibody. Mouse cardiomyocyte cells (HL-1; Gift from Stuart Cook) were cultured in Claycomb medium (Sigma-Aldrich) supplemented with 10% FBS (F2442, Sigma-Aldrich), 10 μg ml−1 penicillin and streptomycin, 2 mM L-glutamine and 0.1 mM norepinephrine. Mouse embryonic fibroblasts were cultured in DMEM supplemented with 10% FCS, 10 μg ml−1 penicillin and streptomycin, and 2 mM L-glutamine.

**Metaphase arrest and propidium iodide staining.** Twenty four hours after passaging the cells, demecolcine (D1925, Sigma-Aldrich) was added to the culture medium, to a final concentration of 0.1 μg ml−1. ESCs and pre-B cells were incubated with demecolcine for 6 h at 37 °C. Fibroblast and cardiomyocyte cells were incubated for 12 h at 37 °C. Cells were collected before and after metaphase arrest (after mitotic shake off). Next, $10^6$ cells were fixed with ice-cold 70% ethanol overnight at −20 °C. Prior to staining, cells were washed twice with PBS and resuspended in staining buffer containing 0.05 mg ml−1 of PI, 1 mg ml−1 RNase A and 0.05% NP40. Samples were incubated for 10 min at room temperature (RT) and 20 min on ice. PI signal was analysed in a linear mode using a BD LSRII flow cytometer and BD DIVA software (version 8.0.1).

**Chromosome preparation and flow sorting.** Chromosomes were prepared using a polyamine-based method[38,82]. Mitotic cells were collected by mitotic shake off. The cells were centrifuged at 289 × g for 5 min at RT. The cell pellets were gently resuspended in 5–10 ml of hypotonic solution (75 mM KCl, 10 mM MgSO4, 0.5 mM spermidine and 0.2 mM spermine; pH = 8) for 15 min at RT. After swelling, cells were then centrifuged at 300 × g for 5 min at RT and resuspended in 1–3 ml of freshly prepared ice-cold polyamine isolation buffer (15 mM Tris-HCl, 2 mM EDTA, 0.5 mM EGTA, 80 mM KCl, 3 mM DTT, 0.25% Triton X-100, 0.2 mM spermine and 0.5 mM spermidine; pH = 7.5). After 15 min of incubation on ice, the chromosomes were released by vortexing vigorously for 20–30 s. To increase chromosome recovery, suspensions were passed through a 22-gauge needle using a 1 ml syringe. Chromosome suspensions were centrifuged for 2 min at 200 × g at RT. The supernatant containing mitotic chromosomes was filtered

using a 20 μm mesh filter into a 15 ml falcon tube. Chromosomes were stained at 4 °C overnight with 5 μg ml−1 Hoechst 33258, 50 μg ml−1 chromomycin A3 and 10 mM MgSO4. At least 1 h prior chromosome sorting, sodium citrate and sodium sulfite were added to chromosome suspensions to a final concentration of 10 mM and 25 mM, respectively. Chromosomes were examined by flow cytometry using a Becton Dickinson Influx (BD FACS software version 1.2.0.142), equipped with spatially separated air-cooled lasers. Hoechst 33258 was excited using a (Spectra Physics Vanguard) 355 nm laser with a power output of 350 mW. Hoechst 33258 fluorescence was collected using a 400 nm long pass filter in combination with a 500 nm short pass filter. Chromomycin A3 was excited using a (Melles Griot) 457 nm laser with a power output of 300 mW. Chromomycin A3 fluorescence was collected using a 500 nm long pass filter in combination with a 600 nm short pass filter. Forward scatter was measured using a (Coherent Sapphire) 488 nm laser with a power output of 200 mW, and this was used as the trigger signal for data collection. Chromosomes were sorted at an event rate of 15,000 per second. A 70 μm nozzle tip was used along with a drop drive frequency set to ~96 kHz and the sheath pressure was set to 65 PSI. Isolated chromosomes were collected in DNA low-binding tubes containing an excess of polyamine buffer.

**Proteomics.** Flow-sorted mitotic chromosomes were pelleted by centrifugation (18,000 × g, 10 min, 5 °C). Supernatant was removed and the obtained pellet was processed by the in-Stage Tip digestion protocol[83] using commercially available iST tips (Preomics, Martinsried, Germany) according to the manufacturer's recommendations. Briefly, pellets were suspended in lysis buffer, and heat denatured, reduced and alkylated on a heated shaking incubator (1,000 r.p.m., 10 min, 95 °C). DNA was fragmented by sonication in an ultrasonic water bath (10 min) and samples were digested with trypsin (500 r.p.m., 1 h, 37 °C). Sample clean-up and desalting was carried out in the iST device using the recommended wash buffers. Peptides were eluted with elution buffer (2 × 100 μl), concentrated in a centrifugal evaporator and resuspended in LC loading buffer (20 μl).

LC-MS/MS analysis was performed as follows. Resuspended protein digests were transferred to auto sampler vials for LC-MS analysis. Peptides were separated using an ultimate 3000 RSLC nano liquid chromatography system (Thermo Scientific) coupled to a Q-Exactive HF-X tandem mass spectrometer (Thermo Scientific) via an EASY-Spray source. Sample volumes were loaded onto a trap column (Acclaim PepMap 100C18, 100 μm × 2 cm) at 8 μl min−1 in 2% acetonitrile and 0.1% TFA. Peptides were eluted on-line to an analytical column (EASY-Spray PepMap C18, 75 μm × 75 cm). Peptides were separated at 200 nl min−1 using a ramped 120 min gradient from 1–42% buffer B in buffer A (buffer A: 5% DMSO and 0.1% formic acid; buffer B: 75% acetonitrile, 0.1% formic acid and 5% DMSO). Eluted peptides were analysed operating in positive polarity using a data-dependent acquisition mode. Ions for fragmentation were determined from an initial MS1 survey scan at 120,000 resolution (at m/z 200) in the orbitrap followed by HCD (higher-energy collisional dissociation) of the top 30 most abundant. MS1 and MS2 scan AGC targets set to 3e6 and 5e4 for a maximum injection time of 25 ms and 50 ms, respectively. A survey scan covering the range of 350–1750 m/z was used, with HCD parameters of isolation width 1.6 m/z and a normalised collision energy of 27%. Data obtained from biological triplicate experiments (each loaded in duplicate) were analysed using the LFQ algorithm in the MaxQuant software platform (version 1.6.2.3)[84], with database searches carried out by the in-built Andromeda search engine against the Swissprot Mus musculus database (16,950 entries, v.20180104). A reverse decoy database was created and results displayed at a 1% FDR for peptide spectrum matches and protein identifications. Search parameters included: trypsin, two missed cleavages, fixed modification of cysteine carbamidomethylation and variable modifications of methionine oxidation, asparagine deamidation and protein N-terminal acetylation. LFQ was enabled with an LFQ minimum ratio count of 2. 'Match between runs' function was used with match and alignment time limits of 0.7 and 20 min, respectively. Statistical analysis as well as data visualisation were performed using the Perseus software platform[85].

Following data processing in MaxQuant, the proteinGroups.txt file was analysed in Perseus (version 1.6.2.2) by uploading the data matrix with the respective LFQ intensities as main columns. The data matrix was reduced by filtering based on categorical columns to remove reverse decoy hits, potential contaminants and protein groups which were 'only identified by site'. Gene Ontology (GO) annotations for taxonomy M. musculus (mainAnnot. mus_musculus.txt) were downloaded from http://annotations.perseus-framework.org. GO annotations for molecular function and cellular compartment were imported by annotating columns based on majority protein IDs. Groups of technical replicate injections and biological replicates of the two conditions ('lysate pellet' and 'flow sorted') were defined in categorical annotation rows. Missing values were replaced with 'NaN' (Quality->Convert to NaN), the technical duplicates were averaged (annot. rows->average groups->mean value; min. one valid value) and data were log transformed (Basic->Transform->log2(x)). Data were then visualised as LFQ intensity histograms (per biological replicate), and LFQ intensity Multi scatter plots and Numeric Venn Diagram numbers were generated in Analysis. Volcano plots were generated based on LFQ intensities with the following settings: test: t-test; side: both; number of randomisations: 250; preserve grouping in randomisations: <none>; FDR: 0.01; S0: 0.1. Hierarchical clustering analysis (HCA) was carried out after filtering rows based on a minimum

of two valid values in at least one group, Z-scoring of values in rows and a two-sample *t*-test of the conditions ('lysate pellet' and 'flow sorted') using the following settings: Student's *t*-test; S0: 0; side: both; FDR: 0.05. After filtering rows retaining *t*-test significant hits only, the HCA was generated with the following settings for both rows tree and columns tree: distance: euclidean; linkage: average; constraint: none; preprocess with *k*-means selected (number of clusters: 300; maximal number of iterations: 10; number of restarts: 1).

**Immunofluorescence on flow-sorted chromosomes.** Flow-sorted chromosomes (chromosomes 19 and X) were spun onto poly-L-lysine-coated slides (VWR) by cytocentrifugation (Cytospin3, Shandon) at 1300 r.p.m. for 10 min at RT. Chromosome samples were blocked with 6% normal goat serum for 1 h at RT and incubated overnight at 4 °C in a humid chamber with primary antibodies to Cenpa (2040 S, Cell Signaling, 1/200), Rad21 (Ab154769, Abcam, 1/100), Myc (SC40, Santa Cruz, 1/200), Sox2 (Ab97959, Abcam, 1/100), Nanog (REC-RCAB0002P-F, 2bScientific, 1/100), Oct4 (sc-5279, Santa Cruz, 1/100), H3K9me3 (07-523, Millipore, 1/200) or H3K27me3 (ab6002, Abcam, 1/200). Chromosomes were washed (buffer containing 10 mM HEPES, 2 mM MgCl$_2$, 100 mM KCl and 5 mM EGTA) and incubated with appropriate secondary antibodies (anti-mouse-Alexa488 (A11001, Invitrogen, 1/200), anti-rabbit-Alexa488 (A11008, Invitrogen, 1/400) or anti-mouse-A566 (A11031, Invitrogen, 1/200) for 1 h at RT. Immuno-stained chromosomes were mounted in Vectorshield mounting medium containing DAPI. Wide-field epi-fluorescence microscopy was performed on an Olympus IX70 inverted microscope using a UPlanApo 100×/1.35 oil objective lens. Super-resolution structured illumination (SIM) microscopy was performed on a Zeiss Elyra microscope using a Plan-Apochromat 63×/1.4 oil objective lens. Fluorescent excitation was performed with 405 nm and 488 nm lasers, and fluorescent emission was collected using bandpass 420–480 nm, bandpass 495–550 nm and long pass 650 nm filters.

**Telomere labelling.** ESCs were transfected with 12 µg of TRF1-YFP plasmid[86]. TRF1-YFP-positive cells were sorted using a BD AriaIII flow sorter and grown in normal ESC medium. Metaphase chromosomes 19 and X from TRF1-YFP-positive cells were sorted, and analysed by optical imaging.

**ATAC-seq.** ATAC-seq was performed in duplicate on asynchronous cells, mitotic cells and purified mitotic chromosomes. For asynchronous cells, the Omni-ATAC-seq protocol was used to obtain nuclei[87]. Briefly, $5 \times 10^4$ ESCs were lysed on ice for 3 min in 50 µl of ATAC-resuspension buffer (10 mM Tris-HCl, pH 7.4; 10 mM NaCl; 3 mM MgCl$_2$) containing 0.1% Igepal CA-630, 0.1% Tween-20 and 0.01% Digitonin. After adding 1 ml of ATAC-resuspension buffer containing 0.1% Tween-20, nuclei were pelleted at $500 \times g$ (10 min at 4 °C). ATAC-seq was subsequently performed largely according to the original protocol[88]. Briefly, nuclei from asynchronous cells, mitotic cells ($5 \times 10^4$) or purified chromosomes ($2 \times 10^6$) were resuspended in 50 µl transposase mixture (25 µl Illumina TD buffer, 22.5 µl H$_2$O and 2.5 µl Illumina TDE1 transposase) and incubated at 37 °C for 30 min with shaking at 1000 r.p.m. After transposition, DNA was purified with the Qiagen MinElute kit and amplified with seven cycles of PCR using the NEBNext High Fidelity master mix and the primers shown in Supplementary Table 1 (ref. [88]). Libraries were subjected to two rounds of Ampure XP bead (Beckman Coulter) purification, including a size selection step using 0.5× beads to remove large fragments. Libraries were assessed by Qubit, Bioanalyzer and with the KAPA Library Quantification Kit (Roche) before sequencing on the Illumina NextSeq system (75 bp, paired end). ATAC-seq data were initially processed using the nfcore/atacseq pipeline version 1.1.0 (ref. [89]; https://doi.org/10.5281/zenodo.2634132), aligning to the mm10 genome to give >45 million mapped read pairs for each library. Mapped reads were imported into Seqmonk (www.bioinformatics.babraham.ac.uk/projects/seqmonk, version 1.46.0) for downstream analysis. Transposase insertion centres were determined by extracting the 5′ ends of all reads (two per read-pair) and offsetting these sites by +4 bp/−5 bp (ref. [88]). For chromosome-wide accessibility profiles, ATAC-seq enrichment was calculated as the number of insertion sites in 25 kb windows, relative to the genome-wide average. For accessibility trend plots, insertion sites were extended ±25 bp to smooth signal and plotted as the average relative distribution across 2 kb windows centred on Esrrb peak summits. Esrrb peak locations and bookmarking status were taken from ref. [16], and coordinates were converted to mm10 using the UCSC (genome.ucsc.edu) LiftOver tool.

**Live cell imaging.** Twenty four hours prior to imaging, Esrrb-tdTomato, Mecp2-eGFP, Pcl2-halo and Suz12-halo-tagged ESCs were grown in ESC medium in Ibidi µ-Sildes 8 Well (Ibidi, 80826), pre-coated with gelatin. Halo-tagged Pcl2 and Suz12 fusion proteins were labelled with 200 nM of Janelia Fluor 549 HaloTag ligand (Promega) for 15 min at 37 °C in 5% CO$_2$. Cells were washed twice with PBS, then washed once with ESC medium for 15 min. A total of 30 min prior to live cell imaging, cells were switched to medium without phenol red (Gibco, 31053-028) and were incubated with 1 µM of SiR-DNA (SC007, SpiroChrome). Time-lapse images were acquired on a Leica TCS SP8 confocal microscope using a 63×/1.40 NA oil objective and Ludin environmental chamber kept at 37 °C with a 5% CO$_2$ supply. Z-stacks were collected every 90 s with a step size of 4 µm. To clearly

visualise all stages of mitosis, individual focal planes were extracted from the z-stacks and a Gaussian Blur (sigma = 1) was applied to the time series in Fiji (version 1.52p) to reduce noise.

**Flow-sorted chromosome size measurements.** After flow sorting, chromosome 19 and chromosome X ($10^5$) were spun onto poly-L-lysine-coated slides by cytocentrifugation (Cytospin3, Shandon) at 1300 r.p.m. for 10 min at RT. Chromosomes were stained with anti-Cenpa (2040S, Cell Signalling, 1/200), washed three times, then incubated with secondary antibody (A11001, Invitrogen). Immuno-stained samples were mounted in Vectorshield mounting medium containing DAPI before optical imaging. Chromosome images were acquired using wide-field epi-fluorescence microscopy performed on an Olympus IX70 inverted microscope (Micro-Manager version 1.4.22) using a UPlanApo 100×/1.35 Oil Objective lens. SIM microscopy was performed on a Zeiss Elyra microscope (ZEN 2012 SP4, version 13.0.2.518) using a Plan-Apochromat 63×/1.4 oil objective lens. Flourescent excitation was performed with 405 nm and 488 nm lasers, and fluorescent emission was collected using bandpass 420–480 nm, bandpass 495–550 nm, and long pass 650 nm filters. Images were analysed using Fiji/ImageJ (version 1.52p) software[90]. Chromosome and centromere size measurements of SIM imaging data were assessed using a custom macro in Fiji to estimate chromosome (total DAPI) and centromere (DAPI high) areas.

**Metaphase spreads.** Exponentially growing cells (60–80% confluent if adherent) were incubated with 0.1 µg ml$^{-1}$ demecolcine solution to arrest cells at metaphase. Adherent cells were then trypsinized and pelleted; non-adherent cells were pelleted directly (5 min at $289 \times g$). Pelleted cells were resuspended in hypotonic solution (40 mM KCl, 0.5 mM EDTA, 20 mM HEPES, pH to 7.4 using NaOH, pre-warmed to 37 °C) for 25 min at 37 °C. Nuclei were pelleted (8 min at $500 \times g$) and supernatant removed (apart from a small drop to re-suspend the pellet) prior to addition of 3:1 MeOH:glacial acetic acid (both Fisher Chemical) fixative (made fresh and pre-cooled to −20 °C) to the top of the tube. The tubes were stored at −20 °C overnight. The next day nuclei were pelleted (8 min at $500 \times g$) and washed in fresh 3:1 MeOH:glacial acetic acid three times before preparation of metaphase spreads. To prepare chromosome spreads nuclei were pelleted and resuspended in a small volume of fixative (to a pale grey solution). A 20 µl drop of 45% acetic acid in water was pipetted onto a glass Twinfrost microscope slide and 23 µl of spread mixture dropped onto it, tilting the slide to spread the nuclei. The slides were air-dried and stored dry at RT. XCyting Mouse Chromosome Painting Probes (Metasystems Probes) to chromosomes X and 19 were used alone or together with mouse gamma satellite probes (DNA a gift from Niall Dillon) directly labelled with FluoroRed (Amersham Life Science RPN2122) by nick translation, to detect chromosomes X or 19 with pericentromeric DNA. Metaphase chromosome painting was performed according to the protocol supplied by Metasystems Probes and mounted in Vectashield containing DAPI. Leica SPII confocal microscope was used for imaging.

**Chromosome painting on flow-sorted chromosomes.** Flow-sorted chromosomes 19 and X ($10^5$) were spun onto poly-L-lysine-coated slides by cytocentrifugation (Cytospin3, Shandon) at 1300 r.p.m. for 10 min at RT. Samples were hybridised with mouse chromosome 19 (D-1419-050-FI, Metasystems Probes) or X paints (D-1420-050-OR, Metasystems Probes) according to the manufacturer's instructions.

**Immunofluorescence on cells.** ESCs were cultured on gelatin-coated glass coverslips, pre-B cells were spun onto poly-L-lysine-coated slides by cytocentrifugation (Cytospin3, Shandon) at 1300 r.p.m. for 10 min. Cells were fixed with 3.7% paraformaldehyde and then permeabilized with 0.1% Triton X-100. After blocking with 1% bovine serum albumin and 10% donkey serum (Sigma), cells were incubated with primary antibodies (listed above) at 4 °C overnight. Finally, cells were labelled with Alexa Fluor-conjugated secondary antibodies and nuclear stained with DAPI. For 5mC immunofluorescence, the cells were incubated with 2 N HCl at RT for 40 min and then neutralised with 0.1 M sodium borate (pH 9.0) for 15 min before the blocking step, and anti-5mC (MABE146, Millipore) was used at 1:2000 dilution. Images were taken with an Olympus IX70 inverted fluorescence microscope at 40× magnification.

**Cellular and nuclear size measurements.** Cells were labelled with PI as described above. Data acquisition was performed on an Amnis image stream flow cytometer and analysed using Amnis IDEAS software (version 6.2). Cells in G1 or in G2/M were discriminated on the basis of DNA content using PI intensity. Brightfield measurements were used to estimate cell size, and refined PI measurements were used to delineate and determine nuclear size.

**TEV protease cleavage.** Total chromosomes ($10^7$) or chromosome 19 ($2 \times 10^5$) were incubated with or without the AcTEV protease (Invitrogen, 10 units) for 4 h at RT with gentle rotation, according to the manufacturer's instructions. Samples were then collected for western blot, or prepared for optical imaging or Cryo-ET.

**Cryo-electron tomography.** Samples for cryo-ET were prepared by mixing 10 µl of flow-sorted chromosome 19 with 1 µl of protein-A conjugated to 10 nm colloidal gold (CMC, Utrecht). A total of 2.5 µl of the mixture was pipetted onto

freshly glow-discharged Quantifoil Cu/Rh R3.5/1 200 mesh grids and plunge frozen into liquid ethane after removal of excess liquid using a Vitrobot Mark IV (FEI). Frozen grids were transferred to and stored in liquid nitrogen until imaging. Tilt series data were collected on a FEI Titan Krios operating at 300 keV, equipped with a Quantum energy filter and a K2 direct electron detector (Gatan) operating in counting mode, using SerialEM software (version 3.6)[91]. Tilt series were collected in two directions starting from 0°, at an unbinned calibrated pixel size of 8.4 Å between ±60° with a 1° increment at 9 μm underfocus. A combined dose of 100 e Å$^{-2}$ was applied over the entire series. Tilt series data were aligned and visualised using IMOD (version 4.7)[92].

**Hi-C on isolated chromosomes.** Hi-C was adapted from ref. [93]. Mitotic cell lysate pellets or sorted chromosomes were cross-linked in 1% formaldehyde for 10 min at RT. Chromatin was digested with 600 units of HindIII overnight at 37 °C with rotation. Digested sticky ends were filled in by DNA polymerase I, Large (Klenow) fragment in the presence of 50 nM biotin-14-dATP, 50 nM dTTP, 50 nM dGTP and 50 nM dCTP for 90 min at 37 °C with rotation. Chromatin fragment ends were ligated with 4000 units of T4 DNA ligase by incubating at RT for 6 h with slow rotation. Proteins were digested with proteinase K and chromatin was reverse cross-linked overnight. RNase A was used to remove RNA after decrosslinking. After isolation with phenol/chloroform/isoamyl alcohol (25:24:1 mixture), DNA was precipitated by sodium acetate/ethanol precipitation. DNA was sheared for 9 min with the Bioruptor sonicator (30 s on and 30 s off per minute, using high power setting). DNA fragments in the range of 300–500 bp were selected with AMPure XP beads. After biotinylated DNA was captured on Dynabeads MyOne Streptavidin T1, DNA ends were repaired in a mixture of T4 polynucleotide kinase, T4 DNA polymerase I and DNA polymerase I, large (Klenow) fragment. dATP was then added to the repaired ends using Klenow Fragment (3′→5′ exo⁻). NEBNext adaptors for Illumina were ligated to the dA-tailed ends. After USER enzyme digestion, NEBNext oligos for Illumina were used for library preparation. A PCR titration was performed to determine the minimal number of PCR cycles (eight cycles in this work). Hi-C libraries were sequenced on an Illumina HiSeq 2500 sequencer to generate 2 × 100 bp paired-end reads for downstream analysis. Hi-C data were mapped and processed using bowtie 2 (version 2.3.2) and HiC-Pro (version 2.7.8) with default settings[94,95]. Raw sequencing data were mapped to the *M. musculus* genome (UCSC assembly mm9, NCBI build 37). PCR duplicates and read pairs that aligned on the same restriction fragment were removed. After converting using the HiC-Pro hicpro2juicebox.sh, valid chromatin contacts were normalised by the KR matrix balancing algorithm and hic files were created using the Juicer Pre (Juicer tools version 0.7.5)[96]. Heatmaps of chromatin interaction matrices were generated and visualised using Juicebox (version 1.6.2)[97].

**Western blot.** A total of 10$^7$ flow-purified chromosomes were incubated with ten units of recombinant TEV protease (Invitrogen) for 4 h at RT. After centrifugation, chromosome pellet was resuspended in 30 μl of cold RIPA buffer (50 mM Tris-HCl (pH 8.8), 150 mM NaCl, 1% Triton X-100, 0.5% sodium deoxycholate, 0.1% SDS, 1 mM EDTA, 3 mM MgCl$_2$ and 1× protease inhibitor cocktail (cOmplete EDTA-free, Roche 11873580001)) supplemented with 1.25 U μl$^{-1}$ Benzonase (Sigma, E1014). Samples were incubated for 20 min at RT, mixed with 30 μl of 2× Laemmli sample buffer (65.8 mM Tris-HCl (pH 6.8), 2.2% SDS, 22.2% glycerol, 0.01% bromophenol blue and 710 mM 2-mercaptoethanol), and denatured at 95 °C for 10 min. Western blots were performed according to standard procedures using Immobilon Block-FL (Millipore WBAVDFL01) as fluorescent blocker, and near infra-red detection was carried out using the LI-COR detection system. The following antibodies and dilutions were used: anti-c-Myc (Santa Cruz Biotechnology sc-40, 1:1000) and anti-Histone H3 (Active Motif 61476, 1:5000).

**Reporting summary.** Further information on research design is available in the Nature Research Reporting Summary linked to this article.

## Data availability

The mass spectrometry proteomics data have been deposited to the ProteomeXchange Consortium via the PRIDE partner repository with the dataset identifier PXD015251. Hi-C data are available from GEO with accession number GSE136681. ATAC-seq data are available from GEO with accession number GSE147552. Previously published Hi-C data were downloaded from GSE82144, biotin-tagged Mecp2 ChIP-seq data were downloaded from GSE39610 and genome-wide DNA methylation data were downloaded from GSE30202. All other relevant data supporting the key findings of this study are available within the article and its Supplementary Information files or from the corresponding author upon reasonable request. The Source data underlying Figs. 2b–e, 3c, d, 4e–g and 5c–e, and Supplementary Figs. 1g, 3c, 4b, c and 5e are provided as a Source data file. A reporting summary for this article is available as a Supplementary Information file.

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

## Acknowledgements

We would like to thank Richard Henderson for his encouragement to begin these studies, C. Tyler-Smith for enabling chromosome sorting, T. Adejumo, R. Maggio, F. Pereira, P. Chana, H. Pallikonda and N. Festuccia for expertise and providing reagents, A. Lisini for reading the manuscript and advice. We thank the LMS/NIHR Imperial Biomedical Research Centre Flow Cytometry Facility, as well as the LMS Genomics and LMS Bioinformatics facilities for support. This work was funded by core support from the Medical Research Council UK to the London Institute of Medical Sciences. R.J.K. was supported by the Wellcome Trust (209400/Z/17/Z) and the European Research Council (681440), and M.K.H. was supported by the Wellcome Trust (109102/Z/15/Z).

## Author contributions

D.D. and A.G.F. conceived and designed the study. D.D. performed most of the experiments, designed the figures and contributed to writing the manuscript. A.D. and K.B. conducted experiments and contributed to writing the manuscript. B.P., H.K., A.-C. K., C.W., A.F., N.V., J.E. and Y.G. conducted experiments. T.A.M.B. and A.K.T. performed EM imaging. J.L., B.L.N., M.K.H., R.J.K. and J.G. provided scientific advice and support. M.M. contributed to study design and writing manuscript. A.G.F. wrote the manuscript and supervised the experiments.

## Competing interests

The authors declare no competing interests.
