## [Peer Review File · Nature Communications]

Reviewers' comments:

Reviewer #1 (Remarks to the Author):

In this manuscript, Djeghloul et al provide a proteomic characterization of mitotic chromatin using an elegant FACS approach to purify native mitotic chromosomes. They identify both new and known proteins associated with mitotic chromosomes in mouse ES cells and correlate the presence of typical repressors associated with heterochromatin to the compaction of the chromosomes. Since ES cells are known to display reduced heterochromatin, they further study the difference in compaction with various somatic cells, and show ES cells present particularly uncompact mitotic chromosomes. Overall, the technique is a good addition to the current state-of-the-art and the data interesting although mainly correlative. Below, I provide specific comments and criticisms that I think should be addressed before publishing a paper that may have the potential to change the way the field of mitotic bookmarking is approached.

1/ The key advantage of the technique is to use unfixed, native chromosomes to describe the proteins that are bound. Indeed, several papers have now shown that fixatives may induce drastic miss-localization of proteins, particularly TFs, during mitosis. However, the authors have not clearly ruled out the potential issues of their technique: how can they ensure a massive loss of proteins is not occurring during this preparation and the sorting? I think it is very important the authors show in their purified chromosomes that the chromatin is maintained in the right configuration. I know they test some structural properties, which is important, but since they give a mitotic bookmarking focus to the paper it should be mandatory they provide the following genome-wide descriptions:

- ATAC-seq to confirm that their procedure is not accompanied by a loss of chromatin accessibility at regulatory elements. The mitotic maintenance of significant accessibility has now been shown by several labs.
- ChIP-seq of proven mitotic bookmarking factors in ES cells, such as Esrrb (Festuccia, 2016) and Tbp (Teves, 2018)

If in their purified chromosomes these features are not fully preserved, then it will be important the authors describe and analyze the limitations of the technique.

2/ The technique is based on comparing mitotic lysates and purified chromosomes. However, this may lead to major confusions for proteins that are partially or largely degraded during mitosis. Hence, the authors should analyze bulk asynchronous cells as well to avoid having false positives based on mitotic ratios exclusively.

3/ Connected to point 1, the authors show a number of stainings for several TFs previously shown to bind mitotic chromatin, such as Sox2. These stainings are not convincing at all, they barely show any signal on the chromosomes, while live imaging showed major enrichment on the chromosomes (Teves, 2016; De Luz, 2016).

4/ My last point regarding the technique itself: the authors should provide a more principled comparison with the paper from Schubeler lab published in Nature Communications (Ginno, 2018): which proteins are differentially identified in both studies? For those proteins, what a live imaging approach would show?

5/ In the experiments using DNMT TKO ES cells, the authors observe, as expected, that chromocenters lose K9me3 and gain K27me3 (although this is not described in the text). In contrast, this peri-centric enrichment for K27me3 is not seen in mitotic chromosomes. The authors should comment on this.

6/ The authors conclude heterochromatic regulators participate to the compaction of mitotic chromosomes, based on the study of knock-outs. However, they do not control for a difference that may already be present in interphase (as they do later in the manuscript when they compare

ES and somatic cells). It is important to understand, I believe, whether the defects of compaction are specific to mitotic chromosomes or to the chromatin in general.

7/ The data on MeCP2 and K9me3 is really puzzling. The authors should show that the mitotic X chromosome really displays lower MeCP2 enrichment than the autosomes, at least using live imaging (a simple transfection would suffice in my opinion) but a mitotic ChIP-seq would be ideal. Then, the observation of high H3K9me3 along the mitotic chromosomal arms in MeCP2 KO cells is outstanding and should definitely be supported by a ChIP-seq.

Reviewer #2 (Remarks to the Author):

In this manuscript, Djeghloul and colleagues aim to identify proteins that convey epigenetic information through cell division. The authors isolated mitotic chromosomes using FACS from embryonic stem cells (ESCs) pharmacologically synchronized to mitosis and performed quantitative LC-MS/MS to detect proteins that are either enriched or depleted from mitotic chromosomes. The authors emphasize that they examine the proteome of the non-crosslinked chromosomes, thus avoiding fixation bias. In total, 614 proteins are found to be enriched on mitotic DNA. GO analysis of enriched proteins shows that these proteins have established roles in chromatin organization, DNA packaging, and chromosome architecture. The authors then focus on the proteins involved in transcriptional repression and maintenance of heterochromatin. Using the *Dnmt1,3a,3b* ^{-/-} and *Eed* ^{-/-} ESCs that abolish the DNA methylation and H3K27me3 respectively, the authors show that loss of these repressive marks results in decompaction of mitotic chromosomes. The authors also show that loss of methyl binding protein *Mecp2* leads to decompaction of mitotic DNA; however, the effect is limited to chromosomes that *Mecp2* is binding to. Next, using the same approach for mitotic chromosome isolation as for their proteomic analysis, the authors demonstrate that mitotic ESC chromosomes are larger than their equivalents isolated from differentiated cells. In the final part of the manuscript, the authors show the mitotic chromosomes isolated using their approach are responsive to biological stimuli. Using a cleavable form of Rad21 (a component of the cohesin complex) in pre-B cells, the authors show that their purified native mitotic chromosomes are susceptible for cohesin cleavage, that results in decondensation of mitotic DNA.

While the topic of the maintenance of epigenetic information through the mitosis is timely and important, the presented manuscript has several issues that question the results presented.

1. The main concern with the presented manuscript is the method used to isolate native mitotic chromosomes. To this reviewer, the complex and multistage process of isolating mitotic DNA used by the authors, which includes polyamine-based chromosome release followed by DNA sorting, likely results in drastic changes of mitotic DNA that could affect the outcome of the study. Images of purified chromosomes resemble a blob (Figure 2-5 and S1-S3) do not look like metaphase chromosomes in cells or obtained from classical techniques such as the chromosomal spread. Furthermore, it is difficult to judge how this process that results in a distinct morphological change to chromosome structure affects the proteins that are bound to mitotic DNA. The authors need to validate the results obtained from the purified mitotic chromosomes using *in vivo* or *in situ* approaches. Furthermore, the subsequent experiments that showing changes in mitotic chromosome size or condensation should be validated either using FISH or by chromosomal spreads.

2. Of note, this is not the first study that examines proteins bound to native mitotic chromosomes. In 2010, Ohta and colleagues performed proteomic analysis of mitotic chromosome proteins in DT40 cells. Surprisingly, this study is neither mentioned nor discussed in the presented manuscript. It is unclear how the current study advances findings presented in the 2010 study, particularly since the current study identified much fewer number of factors binding to mitotic chromosomes. Indeed, this may signify a limitation of the current protocol, which relies on

extensive isolation process.

3. Another issue is that in the current version the manuscript lacks focus. A large part of both introduction and discussion is dedicated to the maintenance of information through mitosis by transcription factors (TF). However, TFs are not the main focus of this manuscript and are only briefly mentioned in Figure 1. Instead, authors investigate the transcriptional repressors that are bound to mitotic chromosomes, which role is not established during mitosis. Authors suggest that these repressors are involved in the maintenance of DNA condensation during mitosis by measuring the size of their purified chromosomes. However, authors do not proceed any further to verify their observations in live cells, or to establish the impact of their observation on either cell division or maintenance of epigenetic information through mitosis. Even more, the possible relevance of their observations is not even adequately discussed. The results section focusing on *Mecp2* is not properly introduced, and it is completely unclear why the authors decided to investigate the distribution of histone H3K9me3 along the mitotic chromosome. Again, there is no attempt to understand the functional relevance of this finding.

Reviewer #3 (Remarks to the Author):

This manuscript identifies factors enriched on mitotic chromosomes, with some follow up experiment to show that key repressive factors condense chromatin. Further, sizes of chromosomes appear dependent on differentiation state. This is an interesting manuscript that will be relevant for many chromatin researchers. I have some comments:

When factors are found to bind mitotic chromatin this does not mean they are associated with their cognate binding sites (they may bind mitotic chromatin, but not sequence-specifically, so that would not be a bookmarking interaction necessarily). Imaging and proteomics will also observe such factors that are non-sequence-specifically bound to chromatin (nicely reviewed in Raccaud and Suter FEBS Lett 2018). It may be good to include mention of this in the manuscript.

For the ESC proteomics: It seems most cells are indeed in G2 or M. The authors need to show how many cells are actually in metaphase. It is possible that the entire population has only few metaphase cells and many G2 cells. When the authors then purify mitotic chromosomes from this culture they will selectively extract these from the subpopulation of metaphase cells, and not from the non-metaphase cells. However, the lysate will have both metaphase and the many G2 (non-mitotic) cells as well (i.e. the B-cell lysate in S5b appears mostly G2, not M as the Hi-C map is dominated by interphase features). This could severely affect the proteomic comparison between mitotic chromosomes and the lysates. Thus, the authors must show that most cells are not only having a G2 DNA content but are in (pro-)metaphase. The authors report finding known chromosome associated factors (*Esrrb*, *Sox2*), while finding that proteins known not to associate in G2 or M (MCM proteins) are depleted. Note that the latter depletion is not evidence that the lysate are good controls for metaphase chromosomes (see point above), as MCM proteins dissociate already in G2.

Do the Hoechst and Chromomycin A3 dyes affect protein-DNA interactions?

Overall the purified chromosomes look very odd. This is not what typical prometaphase chromosomes look like at all. Is this due to the isolation procedure?

The authors measured size of chromosomes and centromeres separately. Did the centromere change volume change particularly when *Dnmt1,3a,3b*, *MeCP2* or *Eed* were deleted? Any specific effects at centromeres as compared to arms?

For the cohesin cleavage experiments: Maybe the centromeres are mostly responsible for size increase, given that cohesin presence on arms is very low. The authors mention that they measure size of centromeres separately. So can the authors report the changes in size of centromeres and the extent to which any such changes can explain the change in whole chromosome size?

The authors state: "native metaphase chromosomes purified by FACS, remain responsive to appropriate biological cues". TEV cleavage of a protein is hardly an appropriate biological cue.

The different size of mitotic chromosomes in ESCs vs differentiated cells is very interesting. I think something similar has been observed in published work from the Heald lab in Xenopus.

Other comments:

Can the authors provide a more detailed explanation about the Mitotic Lysate as control was generated. No or few details mentioned in the Material & Methods.

Can the authors compare proteins are enriched on mitotic chromosomes between cell lines? This may explain size differences, and also serve as another threshold for finding significant proteins.

It is a bit puzzling that cohesin is statistically significantly enriched on mitotic chromosomes given that bulk cohesin is unloaded from chromatin in prophase.

For figure 2 and 3 I would suggest doing normal Mitotic spreads as control, especially for the H3K9me3.

For the chromosome size differences between cell lines, cells have to be arrested with similar time in colcemid as this affects chromosome size. (ESC and pre-B for 6h vs fibro and cardio for 12h).

For Figure 2 and 3 mitotic spreads might tell a lot more about chromosome size as control. This could be used to compare fixed vs unfixed cells, yet ratios should not change.

Supplementary figure 4, except for pre-B cells the size (uM) does not change from G1 to G2. PI intensity is normalized by the cell size so shouldn't the PI intensity double. This seems not to be the case for 3 of 4 cells.

Last, in the methods and materials all the centrifuge speeds are written as rpm without mentioning the centrifuge type, so maybe change it to rcf for reproducibility.

Reviewer #4 (Remarks to the Author):

In this manuscript, Fisher and colleagues study the factors associated with chromosomes. They sorted mitotic chromosomes using flow cytometry and analyzed the proteins co-enriched with the chromosomes. Among enriched proteins many of the expected transcription factors, chromatin modifiers, etc. were found as a positive controls for the experiment. The MS experiments are well described, the data is of high quality and acquired using robust protocols and high standards.

A few minor points to be addressed:

The authors discuss the mitotic bookmarking problem. It is a challenging problem and requires a multi-angle approach to assign or score functional relevance of proteins to mitotic chromosomes. For instance as described in PMID: 24561620. It is good that the manuscript does not make strong claims that proteins enriched in a proteomics screen must be relevant to chromosome biology.

Especially, since it depends on an arbitrary decision how to define significantly enriched proteins. The MaxLFQ algorithm is based on the assumption that most of the proteins do not change between the samples. However, in the study more than 60% of proteins identified in chromosomes enriched sample are noted as being significantly enriched or depleted. Does this have an effect on how to select the criteria for significantly regulated proteins?

In Figure 1c-f the x axis is labeled as "relative intensity". The x axis shows a difference between two conditions in log₂ space or an intensity ratio in a linear space typically noted as enrichment. It looks like that Figures 1d-f are derived from figure 1c by removing most of the points, but the significance cut-off of 1% FDR was not adjusted for individual subsets. One way to emphasize selected proteins would be to make all other proteins marked using transparent symbols and keep the same FDR cut-off for all the plots.

Could you please provide the supplementary figure showing pairwise comparison between biological replicas including correlation coefficients. That would help to assess the reproducibility of the proteomics measurements.

In the supplementary methods description could you please describe the steps performed using Perseus software and which version of the software was used.

Have you identified any proteins in chromosome enriched fraction which were not found in the total lysate? Is there a lot of proteins in the lysate identified only using match between runs functionality?

Have you identified any chromosome associated proteins which are annotated as solely cytoplasmic proteins? That would help to assess the robustness of the enrichment protocol.

A hierarchical clustering is a nice representation of the proteomics data. It would serve a similar function as figures 1c-f, but would allow to explore all potentially similarly behaving protein groups. For example see the publication PMID: 29208753. Please consider adding this.

Response to reviewers

Reviewer #1

In this manuscript, Djeghloul et al provide a proteomic characterization of mitotic chromatin using an elegant FACS approach to purify native mitotic chromosomes. They identify both new and known proteins associated with mitotic chromosomes in mouse ES cells and correlate the presence of typical repressors associated with heterochromatin to the compaction of the chromosomes. Since ES cells are known to display reduced heterochromatin, they further study the difference in compaction with various somatic cells, and show ES cells present particularly uncompact mitotic chromosomes. Overall, the technique is a good addition to the current state-of-the-art and the data interesting although mainly correlative. Below, I provide specific comments and criticisms that I think should be addressed before publishing a paper that may have the potential to change the way the field of mitotic bookmarking is approached.

We were encouraged that the reviewer thinks that that our study 'may have the potential to change the way the field of mitotic bookmarking is approached' and have therefore performed a range of new experiments to address the concerns that have been raised.

1/ The key advantage of the technique is to use unfixed, native chromosomes to describe the proteins that are bound. Indeed, several papers have now shown that fixatives may induce drastic miss-localization of proteins, particularly TFs, during mitosis. However, the authors have not clearly ruled out the potential issues of their technique: how can they ensure a massive loss of proteins is not occurring during this preparation and the sorting?

To address this concern, we have compared proteomic data derived from isolated native mitotic chromosomes with data generated using disuccinimidyl glutarate (DSG) as a protein crosslinker, as described by the Festuccia/Navarro labs (Festuccia et al., Genome Research 29, 250-260). We see a very close correspondence between these two data sets, shown below, dispelling any concern that a widespread loss of proteins has occurred during the isolation of native mitotic chromosomes. Whilst we cannot exclude the possibility that some factors are lost or depleted during sorting, we can be confident that factors detected as enriched are bound to mitotic chromosomes, and we have now validated several of these factors by live-cell imaging (see new Figures 1k and 3a, and supplemental videos 1 and 2). We also draw the referee's attention to our response to point 4; comparing our data to those obtained independently by the Schubeler group (Ginno et al., 2018) shows a good correspondence in the number and identity of proteins detected.

I think it is very important the authors show in their purified chromosomes that the chromatin is maintained in the right configuration. I know they test some structural properties, which is important, but since they give a mitotic bookmarking focus to the paper it should be mandatory they provide the following genome-wide descriptions:

- ATAC-seq to confirm that their procedure is not accompanied by a loss of chromatin accessibility at regulatory elements. The mitotic maintenance of significant accessibility has now been shown by several labs.

We did not intend to convey a singular focus on mitotic bookmarking and have edited the revised text accordingly (see revised introduction).

However, we have performed ATAC-seq on isolated native mitotic chromosomes as requested by the referee, as well as in asynchronous and mitotic samples for comparison. These data showed that global chromatin accessibility patterns are broadly preserved in isolated native mitotic chromosome samples, as illustrated in new Figure 1d (genome-scale accessibility map of mouse chromosome 19). In addition, these data showed that sites bookmarked by factors such as *Esrrb* selectively maintain accessibility in native mitotic chromosome samples (new Figure S1f). These data are consistent with the mitotic ATAC-seq profiles of ESCs that were published previously (Festuccia et al., *Genome Research* 29, 250-260) and provide additional assurance that the isolation procedure that we have used does not result in a significant disruption of chromatin accessibility.

- ChIP-seq of proven mitotic bookmarking factors in ES cells, such as *Esrrb* (Festuccia, 2016) and *Tbp* (Teves, 2018)

As standard ChIP protocols require fixation, we do not think this suggestion is the best way to validate the binding of factors on native metaphase chromosome preps; rather we are currently trying to develop a CUT&RUN approach (which will be novel as applied to native metaphase chromosomes) but this is beyond the scope of the current manuscript. However, since analysis of our ATAC-seq data clearly shows that sites bookmarked by factors such as *Esrrb* remain accessible (new Figure S1f), this is consistent with *Esrrb* (which is enriched in the proteomics analysis) continuing to bookmark flow-isolated mitotic chromosomes.

If in their purified chromosomes these features are not fully preserved, then it will be important the authors describe and analyze the limitations of the technique.

Please see comments above.

2/ The technique is based on comparing mitotic lysates and purified chromosomes. However, this may lead to major confusions for proteins that are partially or largely degraded during mitosis. Hence, the authors should analyze bulk asynchronous cells as well to avoid having false positives based on mitotic ratios exclusively.

The primary aim of this study was to identify proteins that are physically associated with ESC chromosomes during mitosis, rather than performing a quantitative comparison of the proteome between different cell cycle stages. From this perspective, false positives would not be an issue irrespective of protein degradation during mitosis. Other laboratories (including Schubeler, Earnshaw and others) have examined chromatin-associated protein dynamics through the cell cycle, in great detail.

3/ Connected to point 1, the authors show a number of stainings for several TFs previously shown to bind mitotic chromatin, such as *Sox2*. These stainings are not convincing at all, they barely show any signal on the chromosomes, while live imaging showed major enrichment on the chromosomes (Teves, 2016; De Luz, 2016).

The antibody-based staining for TFs that we show is well-controlled and specific (now Figure S1g in the revised manuscript). Proteomic analysis and IF based labelling of native metaphase chromosomes shows a very good agreement; *Sox2* and *Rad21* (both enriched) labelling was evident, in contrast to *Oct4* and *Nanog* (not enriched).

4/ My last point regarding the technique itself: the authors should provide a more principled comparison with the paper from Schubeler lab published in Nature Communications (Ginno, 2018): which proteins are differentially identified in both studies? For those proteins, what a live imaging approach would show?

We have compared our data with that published by the Schubeler lab (Ginno et al., 2018, Nat Commun 9, 4048) as requested by the referee. As shown below, we see a large overlap of the proteins detected on mitotic chromatin in these data sets.

In the Schubeler dataset (analysis shown below) we note that similar protein groups show an enrichment (red) or depletion (blue) on mitotic chromatin as those detected by ourselves (see Figures 1e-j), despite the studies using different approaches and studying different species and cells (human T98G cells versus mouse ESCs).

Live cell imaging of mouse ESCs containing HaloTag Suz12, HaloTag Pcl2 or Mecp2-eGFP, factors detected by Schubeler and ourselves, confirmed that PRC2 and Mecp2 remain bound to chromosomes throughout mitosis (revised Figures 1k and 3a and supplemental videos 1 and 2).

5/ In the experiments using DNMT TKO ES cells, the authors observe, as expected, that chromocenters lose K9me3 and gain K27me3 (although this is not described in the text). In contrast, this peri-centric enrichment for K27me3 is not seen in mitotic chromosomes. The authors should comment on this.

We have now described these findings more clearly in the revised manuscript.

6/ The authors conclude heterochromatic regulators participate to the compaction of mitotic chromosomes, based on the study of knock-outs. However, they do not control for a difference that may already be present in interphase (as they do later in the manuscript when they compare ES and somatic cells). It is important to understand, I believe, whether the defects of compaction are specific to mitotic chromosomes or to the chromatin in general.

We do not claim that reduced chromosome compaction is specific to mitosis – indeed we think this very unlikely, particularly since electron spectroscopic imaging studies of mouse epiblasts have shown that altered chromosome structure is evident in interphase as well as mitotic cells.

What is clear is that the differences in mitotic chromosome compaction between wildtype and mutant ESCs, or between somatic cells and ESCs, occur in the absence of detectable differences in nuclear volume in interphase (G1 or G2) (Figure S4 now includes an analysis of mutant ESCs). We have also performed new experiments to confirm that size differences in sorted native mitotic chromosomes are also evident in conventional metaphase spreads analysed by FISH (see revised Figures 2d, 3d and 4g).

7/ The data on MeCP2 and K9me3 is really puzzling. The authors should show that the mitotic X chromosome really displays lower MeCP2 enrichment than the autosomes, at least using live imaging (a simple transfection would suffice in my opinion) but a mitotic ChIP-seq would be ideal. Then, the observation of high H3K9me3 along the mitotic chromosomal arms in MeCP2 KO cells is outstanding and should definitely be supported by a ChIP-seq.

We agree with the referee that high H3K9me3 along the mitotic chromosome arms of *Mecp2* KO cells is interesting, but to fully understand the mechanisms that account for this will require a dedicated study. This is not within the scope of our current study where we use *Mecp2* KO ESCs to ask whether the reduced chromosome compaction seen in DNMT TKO is the result of lack of CpG methylation *per se*, or could be because interactions with methyl binding proteins such as *Mecp2* are abrogated. Our data show that mitotic chromosome compaction is reduced when CpG methylation is intact but *Mecp2* absent, suggesting that compaction could be due to 5mC-binding proteins rather than 5mC itself.

Additionally, as reviewer 2 specifically questions the relevance of including data on H3K9me3 distribution on *Mecp2* KO mitotic chromosomes (originally Figures 3c and S3d), we have (reluctantly) decided to remove it.

Reviewer # 2

In this manuscript, Djegloul and colleagues aim to identify proteins that convey epigenetic information through cell division. The authors isolated mitotic chromosomes using FACS from embryonic stem cells (ESCs) pharmacologically synchronized to mitosis and performed quantitative LC-MS/MS to detect proteins that are either enriched or depleted from mitotic chromosomes. The authors emphasize that they examine the proteome of the non-crosslinked chromosomes, thus avoiding fixation bias. In total, 614 proteins are found to be enriched on mitotic DNA. GO analysis of enriched proteins shows that these proteins have established roles in chromatin organization, DNA packaging, and chromosome architecture. The authors then focus on the proteins involved in transcriptional repression and maintenance of heterochromatin. Using the *Dnmt1,3a,3b* ^{-/-} and *Eed* ^{-/-} ESCs that abolish the DNA methylation and H3K27me3 respectively, the authors show that loss of these repressive marks results in decompaction of mitotic chromosomes. The authors also show that loss of methyl binding protein *Mecp2* leads to decompaction of mitotic DNA; however, the effect is limited to chromosomes that *Mecp2* is binding to. Next, using the same approach for mitotic chromosome isolation as for their proteomic analysis, the authors demonstrate that mitotic ESC chromosomes are larger than their equivalents isolated from differentiated cells. In the final part of the manuscript, the authors show the mitotic chromosomes isolated using their approach are responsive to biological stimuli. Using a cleavable form of *Rad21* (a component of the cohesin complex) in pre-B cells, the authors show that their purified native mitotic chromosomes are susceptible for cohesin cleavage, that results in decondensation of mitotic DNA.

While the topic of the maintenance of epigenetic information through the mitosis is timely and important, the presented manuscript has several issues that question the results presented.

It is encouraging that the reviewer finds this topic timely and important.

1. The main concern with the presented manuscript is the method used to isolate native mitotic chromosomes. To this reviewer, the complex and multistage process of isolating mitotic DNA used by the authors, which includes polyamine-based chromosome release followed by DNA sorting, likely results in drastic changes of mitotic DNA that could affect the outcome of the study.

Clearly chromosome purification methods have the potential to alter mitotic chromatin and 'lose' proteins. However, there is a very good correspondence between the proteomic data obtained by ourselves and that published recently by the Schubeler group (Ginno et al., 2018 Nat Commun 9, 4048), despite these studies using very different approaches and examining cell types from different species (see below). Regarding the use of polyamine buffer, we note that many similar studies have employed this approach (Uchiyama et al., 2004, Hayashihara et al., 2008, Dey et al., 2009, Ohta et al., 2010). In the Ohta study, which analyses the protein composition of mitotic chicken DT40 cells, we also observed a considerable overlap (1835 hits) in data (see below).

We have also further validated our results by performing new live-cell imaging experiments to confirm mitotic binding of candidate factors (new Figures 1k and 3a, and new supplemental videos 1 and 2); by analysing conventional metaphase spreads (new Figures 2d, 3d and 4g); and by performing ATAC-seq to check chromatin status following chromosome sorting (new Figures 1d and S1f).

Taken together, these results suggest that the reviewers concern that the “multistage process of isolating mitotic DNA used by the authors results in a drastic change of mitotic DNA” is probably unfounded.

Images of purified chromosomes resemble a blob (Figure 2-5 and S1-S3) do not look like metaphase chromosomes in cells or obtained from classical techniques such as the chromosomal spread. Furthermore, it is difficult to judge how this process that results in a distinct morphological change to chromosome structure affects the proteins that are bound to mitotic DNA. The authors need to validate the results obtained from the purified mitotic chromosomes using *in vivo* or *in situ* approaches. Furthermore, the subsequent experiments that showing changes in mitotic chromosome size or condensation should be validated either using FISH or by chromosomal spreads.

These are both very good suggestions and we have now performed *in vivo* analyses (live cell imaging) and *in situ* FISH analysis of conventional metaphase spreads to validate our results.

Live cell imaging of mouse ESCs containing HaloTag Suz12, HaloTag Pcl2 or Mecp2-eGFP (kindly provided by R. Klose and A. Bird labs) confirm that PRC2 and Mecp2 remain chromosome-bound throughout mitosis (these data are shown in revised Figures 1k and 3a and in new supplemental videos 1 and 2).

FISH analysis was performed on conventional metaphase spreads using chromosome 19-specific paint and γ -satellite probe (see revised Figures 2d and 3d). We detected an increase in the size of chromosome 19 on spreads prepared from ESCs lacking DNA methylation (Dnmt1,3a, 3b^{-/-}) or PRC2 activity (Eed^{-/-}) as compared to wild type, consistent with the results observed using native mitotic chromosomes isolated by flow cytometry.

As suggested, we also compared the sizes of chromosome 19 and X in conventional mitotic spreads isolated from ESCs, pre-B cells and fibroblasts. The results shown in revised Figure 4g were consistent with mitotic chromosomes from differentiated cells being more compact than those isolated from ESCs.

With regard to the appearance of mitotic chromosomes shown in Figures 2-5, after isolation the samples appear quite ‘typical’ of pro-metaphase chromosomes, as shown below. However, as immunofluorescence labelling experiments require native chromosomes to be immobilised (usually by cytopinning or air drying) before being processed and examined by microscopy, this may explain the slightly puffy appearance of native chromosomes after immunolabelling.

Native mitotic chromatin sample prepared for Cryo-ET (left) and super-resolution optical microscopy (right).

2. Of note, this is not the first study that examines proteins bound to native mitotic chromosomes. In 2010, Ohta and colleagues performed proteomic analysis of mitotic chromosome proteins in DT40 cells. Surprisingly, this study is neither mentioned nor discussed in the presented manuscript. It is unclear how the current study advances findings presented in the 2010 study, particularly since the current study

identified much fewer number of factors binding to mitotic chromosomes. Indeed, this may signify a limitation of the current protocol, which relies on extensive isolation process.

We actually used the Ohta study (2010) as the basis for our proteomic analysis so I was astounded that this paper was not properly referenced in our manuscript. We have added this citation back and can only apologise for this omission. Regarding the total number of proteins detected on mitotic chromosomes by Ohta (4,029) and by ourselves (3,749) the numbers are actually quite comparable. Although the number of proteins we define as 'enriched' on mitotic ESC chromosomes is lower, most of these have a nuclear annotation (red, see below), with significantly fewer cytoplasmic/plasma membrane proteins (blue, see below). This suggests that our approach of physically isolating and purifying mitotic chromosomes may minimize 'contamination' from interphase cells or other cellular compartments that can confound conventionally-derived mitotic lysate samples.

It is probably not that productive to discuss the advances made in our study versus the Ohta 2010 paper, but one major consideration is that we define the mitotic proteome of mouse ESCs, a widely used and fundamental tool for studies of stem cells and differentiation.

3. Another issue is that in the current version the manuscript lacks focus. A large part of both introduction and discussion is dedicated to the maintenance of information through mitosis by transcription factors (TF). However, TFs are not the main focus of this manuscript and are only briefly mentioned in Figure 1. Instead, authors investigate the transcriptional repressors that are bound to mitotic chromosomes, which role is not established during mitosis. Authors suggest that these repressors are involved in the maintenance of DNA condensation during mitosis by measuring the size of their purified chromosomes. However, authors do not proceed any further to verify their observations in live cells, or to establish the impact of their observation on either cell division or maintenance of epigenetic information through mitosis. Even more, the possible relevance of their observations is not even adequately discussed. The results section focusing on Mecp2 is not properly introduced, and it is completely unclear why the authors decided to investigate the distribution of histone H3K9me3 along the mitotic chromosome. Again, there is no attempt to understand the functional relevance of this finding.

As suggested, we have revised and refocused the introduction (towards transcriptional repressor complexes in ESC mitosis) and now better explain the relevance of our findings.

We have also provided live cell imaging, as requested, to confirm that Polycomb Repressor complexes and Mecp2 (in addition to known transcription factors such as Esrrb) remain bound to chromosomes throughout mitosis (new Figures 1k and 3a and in new supplemental videos 1 and 2). We have also

validated size measurements of native chromosomes using conventional metaphase spreads (new Figures 2d, 3d and 4g).

We have better introduced Mecp2, and have removed information regarding H3K9me3 spreading along the chromosome arms in Mecp2-deficient ESCs (previously Figures 3c and S3d) to better focus this section, as the reviewer thought this unclear (or tangential to the current study).

Reviewer # 3

This manuscript identifies factors enriched on mitotic chromosomes, with some follow up experiment to show that key repressive factors condense chromatin. Further, sizes of chromosomes appear dependent on differentiation state. This is an interesting manuscript that will be relevant for many chromatin researchers. I have some comments:

When factors are found to bind mitotic chromatin this does not mean they are associated with their cognate binding sites (they may bind mitotic chromatin, but not sequence-specifically, so that would not be a bookmarking interaction necessarily). Imaging and proteomics will also observe such factors that are non-sequence-specifically bound to chromatin (nicely reviewed in Raccaud and Suter FEBS Lett 2018). It may be good to include mention of this in the manuscript.

This is an important point – we have cited the Raccaud and Suter paper in the revised manuscript and are more explicit about using proteomics to detect factors bound to mitotic chromatin, rather than factors binding to their cognate sites.

For the ESC proteomics: It seems most cells are indeed in G2 or M. The authors need to show how many cells are actually in metaphase. It is possible that the entire population has only few metaphase cells and many G2 cells. When the authors then purify mitotic chromosomes from this culture they will selectively extract these from the subpopulation of metaphase cells, and not from the non-metaphase cells. However, the lysate will have both metaphase and the many G2 (non-mitotic) cells as well (i.e. the B-cell lysate in S5b appears mostly G2, not M as the Hi-C map is dominated by interphase features). This could severely affect the proteomic comparison between mitotic chromosomes and the lysates. Thus, the authors must show that most cells are not only having a G2 DNA content but are in (pro-)metaphase.

There is a potential misunderstanding here about exactly how these experiments were performed; the mitotic lysates were obtained by polyamine release, and a centrifugation (200 g) step to remove all whole cells, including those in interphase or G2 following metaphase arrest. Mitotic lysate chromatin was then recovered by high speed centrifugation (18,000 g) and this was the starting point for the proteomic analysis. To avoid confusion, we have added this information to revised Figure 1a.

We should also clarify that B-cell lysates (such as those shown in the HiC maps in Figure S5c) were not used as a substrate for proteomic analyses.

Nonetheless, I entirely agree with the reviewer that metaphase-arrested samples often contain interphase contaminants; this was one of the main reasons why we developed a method to physically purify/sort mitotic chromosomes.

The authors report finding known chromosome associated factors (Esrrb, Sox2), while finding that proteins known not to associate in G2 or M (MCM proteins) are depleted. Note that the latter depletion is not evidence that the lysate are good controls for metaphase chromosomes (see point above), as MCM proteins dissociate already in G2.

Agreed – we did not intend to imply that ‘lack’ of MCMs was indicative of mitosis, but rather, that this is what one might anticipate seeing in samples of metaphase chromosomes.

Do the Hoechst and Chromomycin A3 dyes affect protein-DNA interactions?

Hoechst and chromomycin A3 are members of the reversible minor groove binding family that preferentially bind either A/T or G/C tracts, respectively. Chromomycin A3 and Hoechst dyes can be cytotoxic and have been shown to impede both DNA replication and transcription *in vivo* and in cell-free assays (e.g. White et al. 2000, Biochemistry 39; 12262-12273). However, Hoechst 33258 has also been widely-used to image living cells across cell generations (Martin, Leonhardt & Cardoso, 2005 Cytometry A. 67, 45-52). Although we acknowledge that Chromomycin A3 and Hoechst 33258 could displace some factors, we can be confident that factors detected as enriched are bound to mitotic chromosomes. We also note that proteomic comparisons of our data with those obtained independently by the Schubeler lab (in

which neither dye is used) are remarkably similar (see below), suggesting that these dyes do not grossly affect protein recovery. Another indicator that protein binding is largely conserved is that chromatin accessibility, as measured by ATAC-seq, is preserved after chromosome labelling and sorting (new Figures 1d and S1f).

In Ginno et al., 2018 (Schubeler’s dataset) similar protein groups were enriched (red) or depleted (blue) on mitotic chromatin as detected by ourselves (see Figures 1e-j).

Overall the purified chromosomes look very odd. This is not what typical prometaphase chromosomes look like at all. Is this due to the isolation procedure?

After isolation the chromosomes resemble ‘typical’ pro-metaphase chromosomes, as shown below. As immunofluorescence labelling requires these samples to be immobilised on glass slides (by cytospin or air drying), native mitotic chromosomes appear more ‘puffy’ after immuno-labelling.

Native mitotic chromatin sample prepared for Cryo-ET (left) and super-resolution optical microscopy (right).

The authors measured size of chromosomes and centromeres separately. Did the centromere change volume change particularly when Dnmt1,3a,3b, MeCP2 or Eed were deleted? Any specific effects at centromeres as compared to arms?

This is a great suggestion. We have analysed centromere size/volume using DAPI intensity (on native chromosomes) or FISH with probes for γ -satellite (on conventional mitotic spreads). An increase in centromere size was seen in native mitotic chromosomes that lack DNA methylation, PRC2 activity or Mecp2, as shown in revised Figures 2b, 2c and 3c (right-hand panels). Consistent with this we observed an increase in the size of pericentric domains on conventional metaphase chromosome spreads prepared from ESCs that lack DNA methylation or Mecp2 (revised Figures 2d and 3d, right-hand panels).

We have also now examined chromosome 19 and X centromere sizes in the different cell types. Here we found that centromere size remained relatively constant (revised Figures 4e and 4f, right hand panels), an observation that we confirmed using conventional metaphase spreads labelled with γ -satellite probe (Figure 4g, right hand panels).

For the cohesin cleavage experiments: Maybe the centromeres are mostly responsible for size increase, given that cohesin presence on arms is very low. The authors mention that they measure size of centromeres separately. So can the authors report the changes in size of centromeres and the extent to which any such changes can explain the change in whole chromosome size?

This analysis is shown as a new panel in the revised Figure 5d. We do see an increase in the size of chromosome 19 centromeres following TEV cleavage, but this change does not account for the total increase in chromosome size, which also appears to extend into the arms.

The authors state: "native metaphase chromosomes purified by FACS, remain responsive to appropriate biological cues". TEV cleavage of a protein is hardly an appropriate biological cue.

We have rephrased this because, as the reviewer rightly points out, although cohesin cleavage is biologically relevant, the setting and method of cleavage is artificial.

The different size of mitotic chromosomes in ESCs vs differentiated cells is very interesting. I think something similar has been observed in published work from the Heald lab in *Xenopus*.

We are grateful to the reviewer for this 'heads up' on the work of the Heald lab. We have introduced a new paragraph to our revised discussion about factors implicated in regulating mitotic chromosome size, based on work in *Xenopus*, *Drosophila*, *C. elegans* and in mammals.

Previous studies have linked chromosome size and nuclear size (and RanGTP levels), however we observed differentiation-associated changes in mitotic chromosome size in the absence of altered nuclear volume (Figure S4b). We note that several chromosome-associated factors – including CENPA, topoisomerase II, cohesin/condensin, linker histones, core histones and Aurora B kinase – have each been implicated in the regulation of mitotic chromosome size and compaction (Heald and Gibeaux, 2018, Current Opinion in Cell Biology, 52, 88-95) – and these may provide useful leads for future studies.

Other comments:

Can the authors provide a more detailed explanation about the Mitotic Lysate as control was generated. No or few details mentioned in the Material & Methods.

In the revised manuscript (and in our answers above) we explain in greater detail how the mitotic lysates were prepared, and have also added some of this information to Figure 1a.

Can the authors compare proteins are enriched on mitotic chromosomes between cell lines? This may explain size differences, and also serve as another threshold for finding significant proteins.

We have begun to compare proteins bound to mitotic chromosomes in different cell lines. These show a degree of conservation in mitotic chromatin components (as well as anticipated cell type-specific differences) but we believe this is best-explored in a dedicated future study.

It is a bit puzzling that cohesin is statistically significantly enriched on mitotic chromosomes given that bulk cohesin is unloaded from chromatin in prophase.

Although we see that the majority of cohesin is lost from chromosomes in prophase, our data clearly show that a significant amount of cohesin remains associated with centromeric domains later in mitosis (Figures S5a).

For figure 2 and 3 I would suggest doing normal Mitotic spreads as control, especially for the H3K9me3.

We have performed conventional FISH analysis (for chromosomes 19 and X) on mitotic spreads as requested, and these new data are shown in revised Figures 2d, 3d and 4g.

We see increased mitotic chromosome size upon loss of DNA methylation or PRC2 activity, consistent with our observations using isolated native mitotic chromosomes. We did not examine changes in the distribution of H3Kme3 in *Mecp2*^{-/-} ESCs, because one reviewer considered this aspect to be beyond the focus of our current study.

For the chromosome size differences between cell lines, cells have to be arrested with similar time in colcemid as this affects chromosome size. (ESC and pre-B for 6h vs fibro and cardio for 12h).

We very carefully titrated the dose and time of demecolcine treatment for each cell line to obtain the maximal proportion of metaphase-arrested cells. The differing sensitivity of each cell type to mitotic arrest, together with very different cell cycle times, means that this has to be adjusted to reduce heterogeneity between samples.

For Figure 2 and 3 mitotic spreads might tell a lot more about chromosome size as control. This could be used to compare fixed vs unfixed cells, yet ratios should not change.

Please see above answer; we used FISH analysis to examine chromosome 19 and X in mitotic spreads prepared from wildtype ESCs, or cells lacking DNA methylation (*Dmmt1,3a,3b*^{-/-}) or PRC2 activity (*Eed*^{-/-}). The results confirm that mitotic chromosomes lacking these repressive modifications are less compact. These differences were less pronounced than those seen comparing unfixed flow-sorted mitotic chromosomes, something that may reflect the fixation (methanol:acetic acid) and dehydration of samples that occurs in conventional mitotic spread preparation and DNA FISH analysis.

Supplementary figure 4, except for pre-B cells the size (uM) does not change from G1 to G2. PI intensity is normalized by the cell size so shouldn't the PI intensity double. This seems not to be the case for 3 of 4 cells.

Apologies, this was our mistake - the y axis of the bar graphs refers to area and not PI intensity, and this has now been corrected. PI intensity does double from G1 to G2/M (Figure S4a) and was used to distinguish G1 from G2 nuclei. Using the image stream software (IDEAS), brightfield images and PI labelling were subsequently used to delineate and measure cell and nuclear areas respectively.

Last, in the methods and materials all the centrifuge speeds are written as rpm without mentioning the centrifuge type, so maybe change it to rcf for reproducibility.

We have revised this.

Reviewer # 4

In this manuscript, Fisher and colleagues study the factors associated with chromosomes. They sorted mitotic chromosomes using flow cytometry and analyzed the proteins co-enriched with the chromosomes. Among enriched proteins many of the expected transcription factors, chromatin modifiers, etc. were found as a positive controls for the experiment. The MS experiments are well described, the data is of high quality and acquired using robust protocols and high standards.

We were pleased that this referee considers our data to be of high quality and robust. We also thank the referee for their encouragement and constructive suggestions.

A few minor points to be addressed:

The authors discuss the mitotic bookmarking problem. It is a challenging problem and requires a multi-angle approach to assign or score functional relevance of proteins to mitotic chromosomes. For instance as described in PMID: 24561620. It is good that the manuscript does not make strong claims that proteins enriched in a proteomics screen must be relevant to chromosome biology. Especially, since it depends on an arbitrary decision how to define significantly enriched proteins.

We agree with the reviewer's comments. We have also included a citation for the study from Anja Groth's team (PMID: 24561620).

The MaxLFQ algorithm is based on the assumption that most of the proteins do not change between the samples. However, in the study more than 60% of proteins identified in chromosomes enriched sample are noted as being significantly enriched or depleted. Does this have an effect on how to select the criteria for significantly regulated proteins?

The MaxLFQ algorithm exploits a proportion of the proteome which remains unchanged between samples for normalisation. However, this proportion does not have to encompass the majority of protein hits. The original publication introducing the MaxLFQ algorithm (Cox et al., 2014, Mol Cell Proteomics, 13, 2513) used 'spike in' experiments to show the validity of the computational approach. In Fig 3a, a 1:1 mix of *E.coli:H.sapiens* was compared to a 3:1 mix (i.e. all proteins change in abundance). Mean protein ratios remained accurate in the MaxLFQ approach even in this scenario.

In Figure 1c-f the x axis is labeled as "relative intensity". The x axis shows a difference between two conditions in log₂ space or an intensity ratio in a linear space typically noted as enrichment.

We have corrected this.

It looks like that Figures 1d-f are derived from figure 1c by removing most of the points, but the significance cut-off of 1% FDR was not adjusted for individual subsets. One way to emphasize selected proteins would be to make all other proteins marked using transparent symbols and keep the same FDR cut-off for all the plots.

As suggested, we have kept the same cut off for all the plots (1% FDR), emphasized selected proteins and marked other proteins using a transparent symbol.

Could you please provide the supplementary figure showing pairwise comparison between biological replicas including correlation coefficients. That would help to assess the reproducibility of the proteomics measurements.

The pairwise comparison between biological replicates and correlation coefficients have been added to the revised manuscript and are shown in a new supplementary Figure S1c.

In the supplementary methods description could you please describe the steps performed using Perseus software and which version of the software was used.

This information has been added to the revised manuscript.

Have you identified any proteins in chromosome enriched fraction which were not found in the total lysate?

11 protein hits in this category were reliably detected (3/3 replicates), as shown below:

Proteins exclusively detected in flow-purified chromosome fractions.

Protein IDs	Protein names	Gene names	no. of peptides
P35492	Histidine ammonia-lyase	Hal	6
P49962	Signal recognition particle 9 kDa protein	Srp9	3
P52927	High mobility group protein HMGI-C	Hmga2	3
P56581	G/T mismatch-specific thymine DNA glycosylase	Tdg	2
Q08189	Protein-glutamine gamma-glutamyltransferase E	Tgm3	2
Q3U1J1	TCF3 fusion partner homolog	Tfpt	4
Q7TNS8	Uncharacterized protein C17orf96 homolog		2
Q8K015	Centromere protein O	Cenpo	3
Q8VEK6	Inhibitor of growth protein 3	Ing3	4
Q9D084	Centromere protein S	Apitd1	4
Q9EST1;Q32M21;Q5Y4Y6	Gasdermin-A;Gasdermin-A2;Gasdermin-A3	Gsdma;Gsdma2;Gsdma3	2

Is there a lot of proteins in the lysate identified only using match between runs functionality?

There is a subset of 14 protein identifications (after removal of Keratin sequences) where identification by MS/MS occurred in the flow sorted samples and identification in lysate was only supported by 'match-between-runs' (see table below). Two of these protein IDs are also included in the previous table (proteins exclusively detected in flow-purified chromosome fractions). The apparent contradiction is explained by the fact that identification by MS/MS or match-between-runs corresponds to raw protein intensities, while the above table was compiled based on proteins quantified as LFQ intensities.

Majority protein IDs	Protein names	Gene names	Razor + unique peptides	Unique peptides	Q-value	SUM_By MS/MS_lystate	SUM_By MS/MS_sorted	SUM_By matching_lystate	SUM_By matching_sorted
Q8R1F0	Leydig cell tumor 10 kDa protein homolog	D8Erttd738e	2	2	0.00037	0	5	6	1
Q8CHK4	Histone acetyltransferase KAT5	Kat5	3	3	0.009678	0	3	6	3
P97350	Plakophilin-1	Pkp1	4	4	0	0	5	5	1
Q9Z2E1	Methyl-CpG-binding domain protein 2	Mbd2	7	7	0	0	6	5	0
P53568	CCAAT/enhancer-binding protein gamma	Cebpg	1	1	0	0	6	3	0
Q61176	Arginase-1	Arg1	2	2	0	0	6	1	0
P27790	Major centromere autoantigen B	Cenpb	2	2	0.001253	0	6	3	0
Q9EST1;Q32M21;Q5Y4Y6	Gasdermin-A;Gasdermin-A2;Gasdermin-A3	Gsdma;Gsdma2;Gsdma3	2	2	0.002796	0	4	1	2
Q8K015	Centromere protein O	Cenpo	3	3	0	0	6	1	0
Q61233	Plastin-2	Lcp1	2	2	0	0	1	4	2
Q6PD05	Zinc finger protein 821	Znf821	2	2	0.00125	0	3	1	3
Q497V6	Bromo adjacent homology domain-containing 1 protein	Bahd1	3	3	0.00579	0	3	2	3
Q7TMD7	Desmoglein-4	Dsg4	3	3	0.004624	0	2	1	2
Q9D219	B-cell CLL/lymphoma 9 protein	Bcl9	2	2	0.009377	0	3	1	3

Have you identified any chromosome associated proteins which are annotated as solely cytoplasmic proteins? That would help to assess the robustness of the enrichment protocol.

We have examined which proteins have nuclear or solely cytoplasmic/plasma membrane GO term annotations and also compared our results with those of others. As shown below, solely cytoplasmic/plasma membrane proteins (blue) accounted for 34% of hits on mitotic chromosomes (as compared to 35-36% in studies by others). This was reduced to <10% among proteins enriched on mitotic chromosomes, suggesting that our approach is valid for enriching mitotically associated factors.

A hierarchical clustering is a nice representation of the proteomics data. It would serve a similar function as figures 1c-f, but would allow to explore all potentially similarly behaving protein groups. For example see the publication PMID: 29208753. Please consider adding this.

As suggested by the reviewer, we have included hierarchical clustering of the proteomics data as a new Figure S1d.

REVIEWERS' COMMENTS:

Reviewer #1 (Remarks to the Author):

The authors have addressed the most important concerns I had raised. The paper is now substantially improved and, in my opinion, should be published

Reviewer #2 (Remarks to the Author):

After reviewing the revised manuscript, we find that the authors have extensively revised the manuscript to address the major concerns we have raised in the initial review.

1. To our concern that their multistage isolation process might drastically change the composition of mitotic chromosomes, they provided a comparison (reviewer fig only) to two other similar studies, stating that majority of proteins enriched are shared. Further they validated their results by performing live imaging of 4 proteins from their data set (however Esrrb was done previously). They also did an ATAC-seq showing that accessibility is maintained. Furthermore, they performed mitotic chromosome isolation x-lined with DSG and followed by MS and no significant differences are detected between their native and DGS-xlink proteome (reviewer fig only – for reviewer #1). These new experiments have satisfied our initial concern.

2. Furthermore, the authors repeated the measurements of chromosomes and centromeres using FISH and chromosome spreading and got similar results, although centromere size seems to be more controlled than average chromosome size (Fig 2D). They also provide explanation for their “puffy/lobby” look of chromosomes stating that a) immobilization of native chromosome might cause that look b) providing cryo-ET and super-resolution optical microscopy image which shows quite blobby look.

3. They added the missing reference to Ohta 2010 study. They also provided us the figure where they compare localisation of detected proteins and saying that >80% of their enriched proteins are annotated as the nuclear, whereas Ohta et al., had only ~40% of nuclear proteins. This suggest that they are getting more pure chromosomes then previous studies.

4. They revised the manuscript by downplaying bookmarking by TFs and more focusing on maintaining the chromatin condensation and discussing the possibility of maintaining tx programs by repression.

Taken together, the authors addressed our comments in a satisfactory way.

Reviewer #3 (Remarks to the Author):

The authors have done a remarkable job addressing reviewer comments. I have no more comments except two very minor ones:

- 1) The comparison of the author's data to previously published datasets shown in the rebuttal should be added to the supplemental materials.
- 2) The different arrest times to obtain mitotic chromosomes for different cell types remains a confounding factor in chromosome size estimates and this should be mentioned as a caveat.

Reviewer #4 (Remarks to the Author):

I would like to thank the authors for their efforts on improving the manuscript based on reviewer comments. Originally, I had an impression that the manuscript is not focused on mitotic bookmarking as also stated by the authors in the rebuttal letter: „We did not intend to convey a singular focus on mitotic bookmarking and have edited the revised text accordingly (see revised introduction).“ However, it seems that mitotic bookmarking is a very important finding of the study: „It is probably not that productive to discuss the advances made in our study versus the Ohta 2010 paper, but one major consideration is that we define the mitotic proteome of mouse ESCs, a widely used and fundamental tool for studies of stem cells and differentiation.“

I am not convinced whether the permutation based FDR test used in this study is suitable to define „mitotic proteome“. Typically it is used to compare similar states and not very different states as mitotic proteome vs total proteome. In the figure S1C it is clear that mitotic proteome is different compared to total proteome, but defining a criteria which protein is significantly enriched/depleted and in turn is part of the mitotic proteome is not trivial in this experimental setup. I would suggest to avoid strong statements on the definition of "mitotic proteome" since it requires statistically more robust analysis and experimental set up combination.

Regarding the comment on the MaxLFQ algorithm:“ The MaxLFQ algorithm exploits a proportion of the proteome which remains unchanged between samples for normalisation. However, this proportion does not have to encompass the majority of protein hits. The original publication introducing the MaxLFQ algorithm (Cox et al., 2014, Mol Cell Proteomics, 13, 2513) used 'spike in' experiments to show the validity of the computational approach. In Fig 3a, a 1:1 mix of E.coli:H.sapiens was compared to a 3:1 mix (i.e. all proteins change in abundance). Mean protein ratios remained accurate in the MaxLFQ approach even in this scenario.“

I would like to refer to the same publication (Cox et al., 2014, Mol Cell Proteomics, 13, 2513) and quote the authors suggesting that in 3:1 mix the situation is that not all proteins are changing, but based on their estimate only 31% are changing. „MaxLFQ has the prerequisite that a majority population of proteins exists that is not changing between the samples. How big this population needs to be and what the consequences are if the changing population becomes comparable in size to the non-changing one can be seen in the benchmark dataset itself, in which the changing (E. coli) population comprised 31% of the proteins measured in total. MaxLFQ still operated well under these circumstances.“

It is surprising that only 11 proteins were specific to mitotic proteome. I would have expected many more chromatin specific proteins to be identified after removing large fraction of a „background“ proteome as also mentioned by the authors in the rebuttal letter: „purifying mitotic chromosomes may minimize 'contamination'“.

Regarding the figure S1D, I would recommend to use the full protein list for hierarchical clustering instead of only comparing enriched and depleted protein datasets. I believe that most of the information is lost using this pre-filtering step.

I find the comparisons (bar plot and venn diagram) between Gino et. al and this study informative and would suggest to add this to supplementary material. However, the four volcano plots provided in the rebuttal letter lack the description.

Response to reviewers

Reviewer 1.

We are grateful for the substantial improvements to the manuscript that were prompted by the reviewer's critique, and are very pleased that the reviewer recommends publication.

Reviewer 2.

We thank the reviewer for their comments and are pleased that the revised manuscript addresses their concerns in a satisfactory way.

Reviewer 3.

We thank the reviewer for their input in improving our study and are pleased that they feel that we have done a good job in addressing reviewer comments. Regarding the two minor comments

- (1) We now include the comparison of our data to previously published datasets (Venn diagrams and bar charts), as requested, as a new Supplementary Figure 1g.
- (2) We have included a caveat that different arrest times may confound estimates of chromosome size between different cell types (revised manuscript page 9).

Reviewer 4.

(Referee's remarks to the Author are show in blue):

I would like to thank the authors for their efforts on improving the manuscript based on reviewer comments. Originally, I had an impression that the manuscript is not focused on mitotic bookmarking as also stated by the authors in the rebuttal letter: „We did not intend to convey a singular focus on mitotic bookmarking and have edited the revised text accordingly (see revised introduction).“However, it seems that mitotic bookmarking is a very important finding of the study: „It is probably not that productive to discuss the advances made in our study versus the Ohta 2010 paper, but one major consideration is that we define the mitotic proteome of mouse ESCs, a widely used and fundamental tool for studies of stem cells and differentiation.“

I am not convinced whether the permutation based FDR test used in this study is suitable to define „mitotic proteome“. Typically it is used to compare similar states and not very different states as mitotic proteome vs total proteome. In the supplementary figure 1c it is clear that mitotic proteome is different compared to total proteome, but defining a criteria which protein is significantly enriched/depleted and in turn is part of the mitotic proteome is not trivial in this experimental setup. I would suggest to avoid strong statements on the definition of "mitotic proteome" since it requires statistically more robust analysis and experimental set up combination.

We apologise for any confusion, but we think there is a misunderstanding here. We have not attempted to compared the total proteome to the mitotic proteome; neither have we attempted to distinguish the proteome of cells at different cell cycle stages - as several other studies have done this previously, and have done this very well. Instead, we attempt to define proteins within mitotic samples that remain bound to chromosomes. These are, of course, a subset of mitotic proteins. As fixation is thought to remove some of these from chromosomes, our approach has been to isolate unfixed chromosomes and purify these

from within mitotic samples using flow cytometry. The labelling of Figure 1b has been modified to avoid any confusion on this point.

Regarding the comment on the MaxLFQ algorithm: “The MaxLFQ algorithm exploits a proportion of the proteome which remains unchanged between samples for normalisation. However, this proportion does not have to encompass the majority of protein hits. The original publication introducing the MaxLFQ algorithm (Cox et al., 2014, Mol Cell Proteomics, 13, 2513) used ‘spike in’ experiments to show the validity of the computational approach. In Fig 3a, a 1:1 mix of E.coli:H.sapiens was compared to a 3:1 mix (i.e. all proteins change in abundance). Mean protein ratios remained accurate in the MaxLFQ approach even in this scenario.”

I would like to refer to the same publication (Cox et al., 2014, Mol Cell Proteomics, 13, 2513) and quote the authors suggesting that in 3:1 mix the situation is that not all proteins are changing, but based on their estimate only 31% are changing. „MaxLFQ has the prerequisite that a majority population of proteins exists that is not changing between the samples. How big this population needs to be and what the consequences are if the changing population becomes comparable in size to the non-changing one can be seen in the benchmark dataset itself, in which the changing (E. coli) population comprised 31% of the proteins measured in total. MaxLFQ still operated well under these circumstances.“ Reviewer 4 asks how comparable the samples that we examine are. As discussed previously these are mitotic lysates and flow-sorted mitotic chromosome samples. Scatter plots of biological replicate Label Free Quantification (LFQ) intensities were requested previously and included as Supplementary Figure 1c (now Supplementary Figure 1d). The data indicate that comparisons between biological replicates (i.e. the same sample condition) display Pearson correlation coefficients of 0.96 or higher (≥ 0.95 before normalisation), while comparisons between biological replicates of different sample conditions display Pearson correlation coefficients of >0.76 (≥ 0.72 before normalisation). In our view this indicates that the degree of similarity between the conditions is sufficient to justify using the MaxQuant Label-free quantification algorithm (MaxLFQ).

It is surprising that only 11 proteins were specific to mitotic proteome. I would have expected many more chromatin specific proteins to be identified after removing large fraction of a „background“ proteome as also mentioned by the authors in the rebuttal letter: „purifying mitotic chromosomes may minimize ‘contamination’“.

With regards to proteins identified exclusively in flow-sorted chromosome samples, a list of 11 protein hits were provided in response to the reviewer’s question. These only include proteins detected/quantified in all three biological replicates (i.e. in three of three samples). When taking all hits quantified exclusively in the flow sorted condition into account, including those quantified in only two out of three, or one out of three replicates, this number is 232 protein hits (this information can be seen in Supplementary Data file 1).

Regarding the figure S1D, I would recommend to use the full protein list for hierarchical clustering instead of only comparing enriched and depleted protein datasets. I believe that most of the information is lost using this pre-filtering step.

We have now included a heatmap in Supplementary Figure 1e in which the full protein list is used for hierarchical clustering, as requested by the reviewer.

I find the comparisons (bar plot and venn diagram) between Gino et. al and this study informative and would suggest to add this to supplementary material. However, the four volcano plots provided in the rebuttal letter lack the description.

As requested, a comparison with published datasets is now included in the revised manuscript as new Supplementary Figure 1g.